# The FineWeb Datasets: Decanting the Web for the Finest Text Data at Scale

**Guilherme Penedo**    **Hynek Kydlíček**    **Loubna Ben allal**    **Anton Lozhkov**
**Margaret Mitchell**    **Colin Raffel**    **Leandro Von Werra**    **Thomas Wolf**
🤗 Hugging Face

https://huggingface.co/datasets/HuggingFaceFW/fineweb
https://huggingface.co/datasets/HuggingFaceFW/fineweb-edu

## Abstract

The performance of a large language model (LLM) depends heavily on the quality and size of its pretraining dataset. However, the pretraining datasets for state-of-the-art open LLMs like Llama 3 and Mixtral are not publicly available and very little is known about how they were created. In this work, we introduce FineWeb, a 15-trillion token dataset derived from 96 Common Crawl snapshots that produces better-performing LLMs than other open pretraining datasets. To advance the understanding of how best to curate high-quality pretraining datasets, we carefully document and ablate all of the design choices used in FineWeb, including in-depth investigations of deduplication and filtering strategies. In addition, we introduce FineWeb-Edu, a 1.3-trillion token collection of educational text filtered from FineWeb. LLMs pretrained on FineWeb-Edu exhibit dramatically better performance on knowledge- and reasoning-intensive benchmarks like MMLU and ARC. Along with our datasets, we publicly release our data curation codebase and all of the models trained during our ablation experiments.

## 1 Introduction

Large Language Models (LLMs) have quickly become a ubiquitous technology thanks to their ability to competently perform a wide range of text-based tasks. A driving factor in the success of LLMs has been a steady increase in model sizes [1–3], which in turn necessitate ever-larger pretraining datasets. Beyond scale, other characteristics of pretraining data have proven to be important, including filtering out "low-quality" content [2, 4] and removing duplicate text [5]. Ultimately, the curation choices made when developing a pretraining dataset can have a huge impact on the downstream capabilities and performance of an LLM. As such, pretraining dataset curation strategies are often treated as closely guarded trade secrets. In fact, there are many popular "open" language models whose parameters are publicly available but whose pretraining datasets were not released and are scarcely documented [6, 7]. The lack of access to high-quality large-scale pretraining datasets and lack of information about their curation has led to concerns of a growing gap between proprietary and public knowledge.

In this work, we aim to minimize this gap by developing and releasing the FineWeb datasets, a collection of large-scale pretraining datasets that can be used to train performant LLMs. Specifically, we first introduce FineWeb, a 15-trillion token dataset of text sourced from 96 Common Crawl snapshots. FineWeb is sufficiently large to train a Chinchilla-optimal model [1] with more than 500 billion parameters. Beyond scale, FineWeb's recipe involves a principled strategy for choosing and tuning filtering heuristics that helped produce a small set of effective filters out of over fifty candidate filters from past work. In addition, we performed an in-depth exploration of how different deduplication strategies and granularities can impact performance. To validate our design choices, we ultimately demonstrate that models trained on FineWeb perform better than those trained on other

38th Conference on Neural Information Processing Systems (NeurIPS 2024) Track on Datasets and Benchmarks.

public web-based pre-training datasets. Inspired by recent work advocating for training LLMs on educational data [8, 9], we additionally introduce FineWeb-Edu, a subset of 1.3 trillion tokens from FineWeb that was rated as highly educational by a custom classifier. Models trained on FineWeb-Edu exhibit significantly better performance on knowledge- and reasoning-intensive benchmarks like MMLU [10] and ARC [11]. Both datasets are released under the permissive ODC-By License. Apart from contributing datasets, we also release `datatrove` [12], the data processing library we developed to create FineWeb. On the whole, our work represents a significant step towards improving public knowledge and resources for curating LLM pre-training datasets.

## 2 Background

In this work, we focus on the curation of training datasets for autoregressive Transformer-based large language models (LLMs) [13]. At their core, LLMs aim to produce a distribution over the next token of text conditioned on past tokens, where each token is typically a word or subword unit [3]. The generality of this paradigm allows LLMs to be applied to virtually any text-based task by formulating a prefix whose continuation corresponds to performing the task (e.g. "The cat sat on the mat translated to French is..." for English-to-French translation). Such models may undergo many stages of training including pretraining on unstructured text data, fine-tuning to improve performance on a specific task [14], multitask fine-tuning to improve generalization to new tasks [15], and learning from human feedback to improve instruction-following capabilities [16, 2]. In this work, we focus solely on curating data for the pretraining stage.

While many sources have been considered for pretraining data including text from books [2, 3, 17, 4], Wikipedia [2, 3, 17, 4], and research papers [2, 4, 18], a highly common choice is to use web text, i.e. text scraped from webpages on the public internet [19, 20]. While some companies like OpenAI [21] and Anthropic [22] perform their own web scrapes, designing, implementing, and running a web scraper at scale requires significant resources and expertise. Many LLM pretraining datasets have therefore been constructed from text from the Common Crawl [23], a publicly available and continually updated collection of website snapshots that has been running since 2007. As of writing, Common Crawl has produced 100 web snapshots totaling petabytes of data.

Although Common Crawl has produced more than enough data to train recent LLMs, it has been shown that the performance of an LLM can heavily depend on how web text has been filtered and preprocessed before being used for pretraining [19]. In particular, web text can contain a large amount of "unnatural" language (e.g. "boilerplate" text, gibberish, etc.). Training on unnatural language data can harm the performance of LLMs, possibly because most downstream uses of LLMs do not involve such data. On the other hand, filtering out *too* much content can produce a dataset that is too small to perform sufficient pretraining (which typically involves only one pass or a few passes over the pretraining dataset [24]) for a general use model. Separately, web text can contain a large amount of duplicated content, which has also been shown to be harmful in the context of pretraining data [5]. While deduplication may seem as straightforward as "removing duplicate text", in practice many design choices must be made (line, paragraph, or document-level deduplication? fuzzy or exact matching? etc.). The curation and relative performance of different web text-based pretraining datasets therefore heavily depends on a given dataset's filtering and deduplication pipeline.

Given that the focus of our work is to carefully design an effective Common Crawl-based pre-training dataset, we now briefly discuss the filtering and deduplication used in past public datasets. **OSCAR** [25] processes Common Crawl using a pipeline inspired by that of Touvron et al. [2], which uses a fastText-based language classifier [26] to filter pages based on their language and then performs deduplication at the line level using a non-cryptographic hash algorithm. **C4** [15] uses `langdetect` [27] to filter out non-English pages, then applies a series of heuristic filters (retaining only those lines that end in a terminal punctuation mark, removing short lines, discarding any page that contains a word from a "bad words" list, etc.), and finally deduplicates over three-line windows. **CC-100** [28] uses the `cc_net` pipeline, which includes fastText for language identification, performs paragraph-level hash-based deduplication, and retains only the text that is assigned a low perplexity by a n-gram language model trained on Wikipedia. **The Pile** [29] is a composite dataset that includes "Pile-CC", a collection of text from one Common Crawl snapshot that uses pycld2 [30] for language detection, jusText [31] for boilerplate removal, a classifier to filter out pages that are dissimilar form WebText (described below), and fuzzy deduplication using MinHash [32]. **ROOTS** [33] includes text from the pre-processed web crawl OSCAR with additional heuristic-based filtering and SimHash-

based deduplication. **RedPajama** [34] is a composite dataset that includes Common Crawl-sourced text processed using the `cc_net` pipeline as well as quality filtering using a classifier trained to distinguish Wikipedia level content from random Common Crawl samples. **SlimPajama** [35] further processed RedPajama by removing short documents and performing additional fuzzy MinHash-based deduplication. **RefinedWeb** [36] uses `trafilatura` [37] for text extraction, fastText for language identification, heuristic rules inspired by MassiveText (discussed below) to filter data, and both MinHash (fuzzy) and ExactSubstr (exact) deduplication. **RedPajama v2** [34] has 84 Common Crawl snapshots released unfiltered and non-deduplicated but with labels from filtering techniques from `cc_net`, C4, MassiveText, RefinedWeb and others, as well as deduplication labels for exact (Bloom filter) and fuzzy (MinHash) deduplication. Finally, **Dolma** [38] has a Common Crawl-based subset that uses fastText for language classification, heuristic rules from MassiveText and C4 for quality filtering, rules- and classifier-based toxicity filtering, and URL, document and paragraph-level deduplication using a Bloom filter.

Apart from public datasets, the technical reports accompanying the announcement of closed LLMs occasionally discuss pretraining datasets. **WebText** [20] (used to train GPT-2) involves only those non-Wikipedia webpages that were linked to from Reddit posts with at least 3 karma, with text extracted using Dragnet [39] and Newspaper1 [40] and an unspecified deduplication pipeline. **GPT-3's Dataset** [3] includes content from Common Crawl that has been filtered using a classifier trained on WebText, Wikipedia, and Books, and deduplicated using MinHash. **MassiveText** [41] (used to train Gopher) is a web-based dataset using Google's SafeSearch to remove explicit content and heuristic filters based on document's content (number of words, stop-words appearance, character repetition, etc.) as well as MinHash-based deduplication.

## 3 Building FineWeb

Our design of FineWeb is primarily empirical: we performed a series of "data ablation" experiments to test different methods at each stage of the pipeline. In this section, we chronicle our experimental results and design choices. All ablations follow our iterative dataset building process, i.e., the baseline model for each subsection includes only the processing steps from the previous subsections, unless explicitly mentioned. Our results are fully reproducible with code in our `datatrove` repository.

### 3.1 Experimental setup

We compare pipeline design choices at each stage by training data ablation models that are identical apart from the data they were trained on (same number of parameters, architecture hyper-parameters, and trained on an equal number of randomly sampled tokens from each version of the data). We then evaluated them on the same set of downstream task benchmark datasets (discussed below). To minimize the impact of random data subset selection on evaluation scores, we trained two models for each dataset version, each using a different but equal-sized random subset of the full data and a different initialization seed, and then compared average scores.

All training was performed using the `nanotron` library. Data ablation models all had 1.71B parameters (including embeddings), used the Llama architecture [2] with a sequence length of 2048, a global batch size of ~2 million tokens, and the GPT2 tokenizer [20]. Within a given experiment, all models were trained on the same amount of data for the same number of steps. Filtering ablations were trained on ~28 billion tokens (roughly the Chinchilla-optimal training size for this model size [1]), while some deduplication ablations and runs to confirm cumulative relative performance improvements after each step of filtering were conducted on 350 billion tokens. The full training hyperparameter are available in Appendix D. We make all models trained for our ablations publicly available on our dataset repository. In total, we trained over 70 models on our internal cluster, for an estimated total of 80,000 H100 GPU hours.

Evaluation was performed using the `lighteval` library. We aimed to select a set of benchmarks that would provide good signal at the relatively small scale of our data ablations. Specifically, we chose benchmarks where models showed minimal score variance between runs trained on different random samples of the same dataset; monotonic (or nearly monotonic) score improvement over a given training; and scores above random baseline for models of this size. These criteria ensure that the scores obtained on a subset of the data are representative of the entire dataset and that they reflect a reliable measurement of the effect of the training data on model performance. Ultimately,

we selected benchmark datasets CommonSense QA [42], HellaSwag [43], OpenBook QA [44], PIQA [45], SIQA [46], WinoGrande [47], ARC [11], and MMLU [10], truncating large benchmarks to 1000 samples so that we could efficiently evaluate over the course of training. We publicly release our exact evaluation setup.

## 3.2 Text extraction

Common Crawl data is available in two different formats: WARC and WET. WARC (Web ARChive format) files contain the raw data from the crawl, including the full page HTML and request metadata. WET (WARC Encapsulated Text) files provide a text-only version of crawled websites by using `htmlparser` [48]. While WET files are commonly used as a starting point for dataset creation, similarly to Gao et al. [29], we found that WET files retained too much boilerplate and menu text. We therefore experimented with extracting the text content from the WARC files using the open source `trafilatura` library [49], which from visual inspection of the results provided good quality extraction when compared to other available libraries (less boilerplate and menu text). Custom text extraction is relatively costly, but its effects are felt on model performance: Fig. 1 shows the performance of ablation models trained on either trafilatura applied to WARC data or WET data, with minimal additional filtering (fastText language identification to filter samples with English as the highest probability label) and no deduplication. Using trafilatura-extracted text from WARC files clearly results in a more performant model and we therefore use WARC-based data in all of our following experiments.

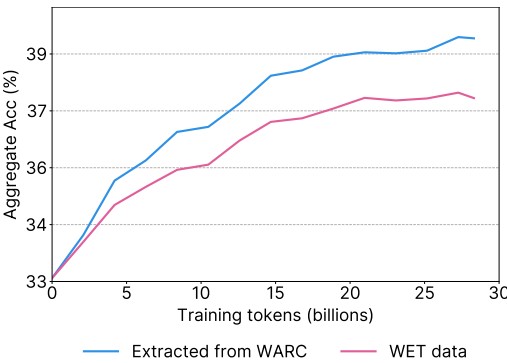

Figure 1: **Trafilatura-extracted WARC vs WET** 28B tokens ablation study. Custom text extraction outperforms the default WET data. No filtering or deduplication was applied except fastText English language filtering.

Figure 2: **Base filtered WARC vs Unfiltered WARC data** 28B tokens ablation study. Our base filtering step provides a significant performance uplift.

## 3.3 Base filtering

As a starting point to our filtering, we applied a basic filtering pipeline using part of the setup from RefinedWeb [50]. Concretely, we applied URL filtering using a blocklist [51] to remove adult content, applied a fastText language classifier [52, 26] to keep only English text with a score $>= 0.65$, and applied quality and repetition filters from MassiveText [41], using the original thresholds. After applying this filtering to all of the WARC-based text extracted from the 96 snapshots available at the time of writing, we obtained roughly 36 trillion tokens of data when tokenized with the GPT-2 tokenizer. Applying these steps results in a performance uplift, as seen in Fig. 2.

## 3.4 Deduplication

The web has many aggregators, mirrors, templated pages or just otherwise repeated content spread over different domains and webpages. Removing these duplicates (deduplicating) has been correlated with improvements in model performance [5] and a reduction in memorization of pretraining data [53, 54]. There are different ways to identify and even define duplicated data. Common approaches rely on hashing techniques or efficient data structures like suffix arrays [55]. Methods can also be "fuzzy" by using a similarity metric or "exact" by checking for exact matches between two text chunks [56].

Following RefinedWeb [50], we experimented with MinHash, a fuzzy hash-based deduplication technique that scales efficiently to many CPU nodes and allows tuning of similarity thresholds (by controlling the number and the number of hashes per bucket) as well as the length of the subsequences considered (by controlling the n-gram size). We chose to collect each document's 5-grams, obtained using an English word tokenizer [57], and computed MinHashes using 112 hash functions in total, split into 14 buckets of 8 hashes each — targeting documents that are at least 75% similar. Documents with the same 8 MinHashes in any bucket are considered duplicates of each other. We then perform a transitive clustering step where documents A, B and C will be in the same duplicate cluster if A and C are duplicates and B and C are duplicates, even if A and B do not have 8 matching MinHashes in any bucket with each other. One (randomly chosen) document is kept per duplicate cluster while the remaining duplicates are removed. We further discuss deduplication parameters in Appendix E.1.

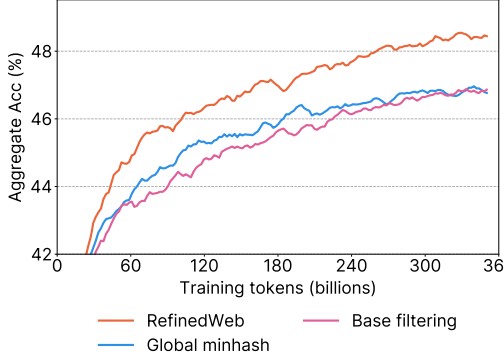
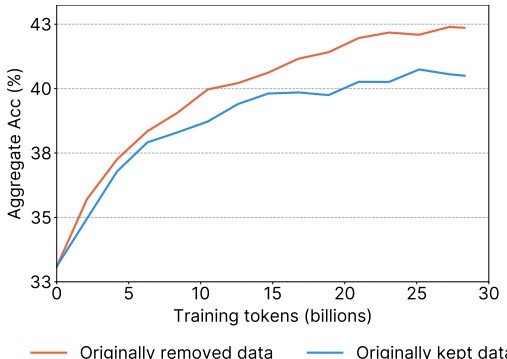

Figure 3: **Global minhash deduplication study**. Applying minhash deduplication globally to the dataset provides only a modest performance uplift, with the resulting model far behind one trained on Refined-Web.

Figure 4: **2013-48 global minhash impact study**. Global deduplication upsamples lower-quality data in the last deduplicated crawl, resulting in worse performance of the retained data compared to the removed data.

Our first approach was to apply MinHash deduplication globally to the entire dataset (all 96 snapshots). We did this in an iterative manner: starting with the most recent snapshot (2023-50, at the time the experiment was run) and proceeding chronologically until we reached the oldest snapshot. When applied to the oldest snapshots, this process removed as much as 90% of the original base filtered data, as they were deduplicated against a large number of other snapshots. Deduplicating the entire dataset in this manner resulted in 4 trillion tokens of data. However, when training on a randomly sampled 350 billion tokens subset, our ablation models showed little improvement over a model trained on the non-deduplicated data, scoring far below RefinedWeb on our aggregate of tasks, as shown in Fig. 3.

This challenged our initial assumption that global deduplication would inevitably result in higher benchmark scores. We therefore performed an additional experiment to investigate the quality of the remaining data. We trained two models on two slices from the older 2013-48 snapshot: (a) the fully deduplicated remaining ~31 billion tokens (*originally kept data*); and (b) 171 billion tokens obtained by individually deduplicating (without considering the other crawls) the ~460 billion tokens that had been removed from this crawl in the iterative deduplication process (*originally removed data*). Results are presented in Fig. 4. They show that, for this older crawl *taken in isolation*, the data from it that was kept (10% of the original data) was actually *of worse quality* than the 90% of data that was removed. We confirmed this by visual inspection: *originally kept data* contains more ads, incoherent lists of keywords and generally badly formatted text than *originally removed data*.

We therefore tried an alternative approach: individually deduplicating each snapshot (independently from the others), using the same parameters as before. This resulted in 20 trillion tokens of data. When training on a random sample from this dataset (with data sampled from all snapshots) it matched RefinedWeb's performance, as per Fig. 5.

One of our hypotheses is that the main improvement gained from deduplication lies in the removal of large clusters of duplicates with *hundreds of thousands* of documents [50] present in all crawls, while further deduplication of clusters with a small number of duplicates (less than ~100, i.e., the number

of crawls) can harm performance. More specific filtering targeting the long tail of data quality might be more suited than deduplication for this subset of the data.

To attempt to improve upon individual deduplication of each snapshot, we additionally experimented with "lighter" global deduplication techniques. Ultimately, none of these techniques improved performance over independent per-snapshot deduplication. A full description of these methods and results are in Appendix E.3.

### 3.5 Adding C4's filters

By this point we had reached the same performance as RefinedWeb [50] using our base filtering and independent MinHash. However, we noted that the C4 dataset [15], while smaller than FineWeb, still showed stronger performance on some of the benchmarks in our evaluation suite, in particular HellaSwag [43], one of the benchmarks in our aggregate group of tasks with the highest signal-to-noise ratio. Despite being one of the first large scale LLM training datasets, C4 is still fre-

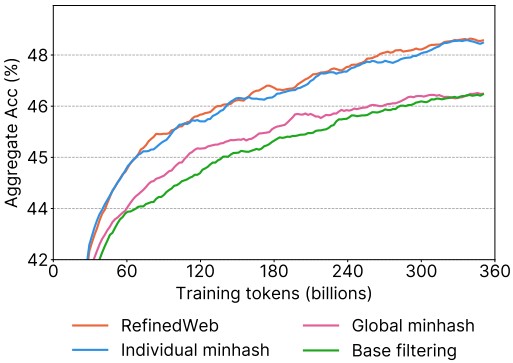

Figure 5: **Individual minhash deduplication study**. Unlike Global minhash, deduplicating individually improves the average score.

quently part of the pretraining data mixture of recent models such as LlamaA 1 [2]. We set out to explore additional filtering steps that would allow us to match or surpass the performance of C4. A natural starting point was to look into the processing of C4 itself.

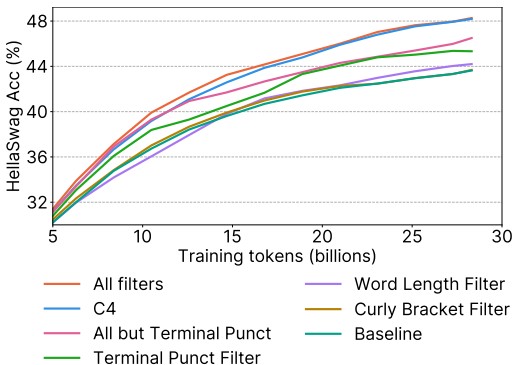

Figure 6: **Comparison of C4 filters impact on HellaSwag benchmark performance**. The Terminal Punctuation filter provides the most significant performance uplift.

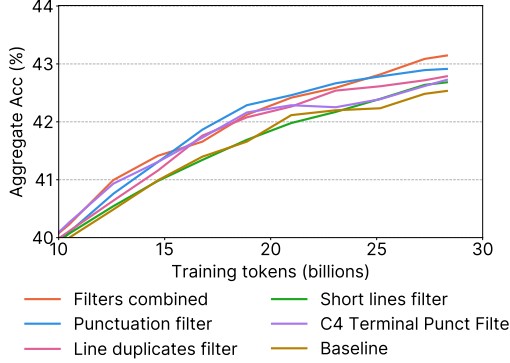

Figure 7: **Custom FineWeb filters study**. The combined filters outperform both the base filtered baseline as well as the best performing C4 filter (Terminal Punctation), while removing less.

C4 was constructed from the 2019-18 crawl by applying heuristic filters, which included dropping lines without a terminal punctuation mark, that mentioned javascript, or that had "terms-of-use"/"cookie policy" statements, and dropping documents that were too short or that contained "lorem ipsum" or a curly bracket ({). We experimented with applying these filters to a baseline of the base filtered and individually deduplicated 2019-18 crawl, and, additionally, compared the results to C4 itself.

Fig. 6 shows that applying *All filters* allows us to match *C4*'s HellaSwag performance; the *Curly bracket filter*, and the *Word lengths filter* only give a small boost, removing 2.8% and 4.3% of tokens, respectively; the *Terminal punctuation filter*, by itself, gives the biggest individual boost, but removes *around 30%* of all tokens; the lorem_ipsum, javascript and policy rules each remove <0.5% of training tokens, so we did not train on them individually; *All but terminal punct* performs better than terminal_punct by itself, while removing less in total (~7%). We decided to apply all C4 filters mentioned above except the terminal punctuation filter, as it eliminates an excessively large amount of data.

### 3.6 Developing additional heuristic filters

Past work has mainly developed heuristic filters through data inspection [15]. In this work we devised a more systematic process for designing heuristic filters and tuning their thresholds. We started by collecting over **50** high-level statistics ranging from document-level metrics (e.g. number of lines, avg. line/word length, etc) to inter-document repetition metrics (inspired by MassiveText [1]) on both a high- and low-quality web dataset. Specifically, we used the individually and globally deduplicated versions of the 2013-48 snapshot (previously mentioned in Section 3.4) as our "high-quality" and "low-quality" datasets respectively. We then identified metrics for which the distribution of values differed significantly across the two datasets, inspected the histograms of the two distributions and empirically chose thresholds that would target sections of the histogram where the lower quality dataset frequency was higher than on the corresponding higher quality dataset section. As an example, we plot the distribution of the *fraction of lines ending with punctuation* metric in Fig. 8. We can see that the higher quality dataset has in general higher document density for larger values of our metric, and, in particular, the lower quality dataset has a much higher density of documents for values $< 0.12$. We thus conclude that documents with a fraction of lines ending with punctuation $< 0.12$ are generally lower quality and use this value as a tentative threshold to filter documents. Following this process for all metrics yielded **16** candidate metric-threshold pairs.

We then assessed the effectiveness of these 16 newly created filters by conducting several 28B token ablation runs on the *2019-18 crawl*. Full details for these runs are in Appendix E.4. Out of all those runs, we identified **three** filters (see their ablations runs in Fig. 7) that demonstrated the most significant improvements on the aggregate benchmark score. Specifically, the chosen filters remove documents where the fraction of lines ending with punctuation is $<= 0.12$ (10.14% of tokens removed vs. 30% from the original C4 terminal punctuation filter), where the fraction of characters in duplicated lines is $>= 0.1$ (12.47% of tokens removed; the original MassiveText threshold for this ratio is $>= 0.2$), and/or where the fraction of lines shorter than 30 characters is $>= 0.67$ (3.73% of tokens removed). When applying the three together, ~22% of tokens were removed and the aggregate score increased by about 1% in the 28B token ablations. These filters allowed us to further improve performance and, notably, surpass the C4 dataset performance while filtering out a smaller proportion of data.

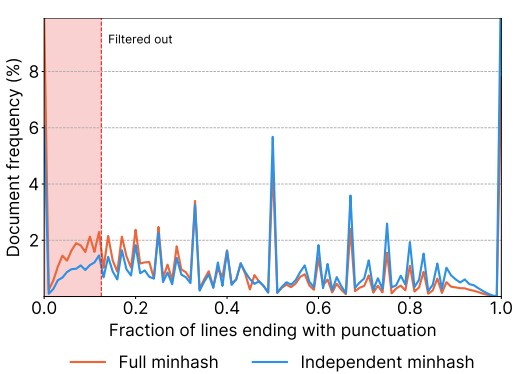

Figure 8: **Impact of Deduplication Methods on 2013-48 Crawl.** Histogram showcasing higher frequency of documents with small fraction of *lines ending with a terminal mark* for Global minhash compared to Indivudal one. A threshold selected for filtering is also indicated.

### 3.7 The final FineWeb dataset

Combining the decisions made in the previous sections and applying the resulting pipeline to 96 Common Crawl snapshots produces the 15T-token FineWeb dataset. Specifically, we extract text from WARC files (Section 3.2), apply base filtering (Section 3.3), perform individual per-crawl MinHash deduplication (Section 3.4), apply a selection of C4 filters (Section 3.5), and finally apply custom filters (Section 3.6). Each step provides a relative performance boost on our group of benchmark tasks, as seen in Fig. 9. For the public release of the dataset, we have also applied Personal Identifiable Information (PII) removal, by anonymizing email and public IP addresses.

In Fig. 10 we compare FineWeb with the following commonly used openly accessible web-scale datasets: RefinedWeb (500B tokens) [50], C4 (172B tokens) [15]; the Common Crawl-based part of Dolma 1.6 (3T tokens) and 1.7 (1.2T tokens) [58], The Pile (340B tokens) [29], SlimPajama (627B tokens) [35], the deduplicated variant of RedPajama2[1] (20T tokens) [34], English CommonCrawl section of Matrix (1.3T tokens) [59], English CC-100 (70B tokens) [60], and Colossal-OSCAR (850B tokens) [61]. Notably, FineWeb shows strong performance and FineWeb-Edu (detailed below) outperforms all other open datasets we compared on our aggregate group of tasks, further validating

---

[1]RedPajama2 includes 40+ quality annotations but is only actually filtered with the CCNet pipeline[28]

the design choices we made. Note that to train these models we randomly sampled 350 billion tokens from each dataset, without upsampling any individual Common Crawl snapshot.

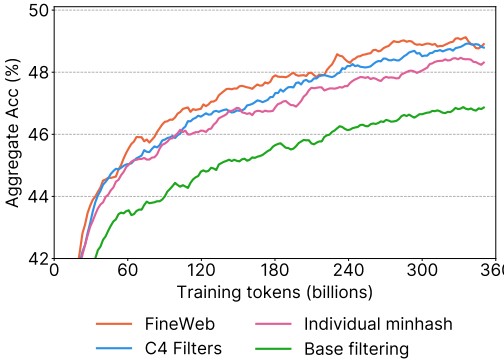

Figure 9: **Each processing step in FineWeb provides a performance uplift.** Compared to the base filtering (Section 3.3), applying individual-crawl Min-Hash deduplication (Section 3.4) the C4 filters (Section 3.5), and our additional heuristic filters (Section 3.6) each improve performance.

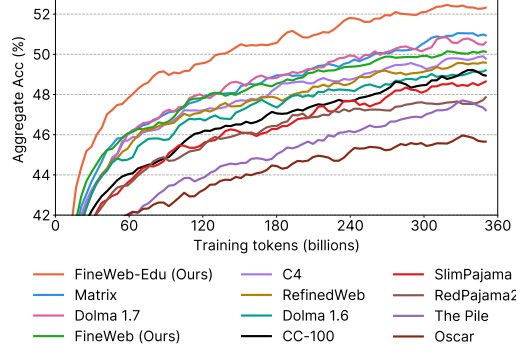

Figure 10: **Comparing FineWeb datasets to other public datasets.** Base FineWeb shows strong performance, with the educational subset (FineWeb-Edu) surpassing all other public datasets and further enhancing the aggregate score by approximately 2%.

## 4   FineWeb-Edu

An interesting approach has recently emerged for filtering LLM training datasets: using synthetic data to develop classifiers for identifying educational content. This technique was notably used in the non-public pretraining datasets of Llama 3 [6] and Phi-3 [8], but its large-scale impact on web data filtering has not been publicly explored. We applied this technique to FineWeb by filtering it with an educational quality classifier developed from synthetic annotations generated by Llama-3-70B-Instruct [62]. The resulting dataset, **FineWeb-Edu**, contains 1.3 trillion tokens. FineWeb-Edu is specifically optimized for educational content and outperforms all openly accessible web-based datasets on a number of reasoning- and knowledge-intensive benchmarks such as MMLU, ARC, and OpenBookQA by a significant margin.

To build the synthetic annotations, we use Llama-3-70B-Instruct to score 460,000 randomly sampled webpages from the FineWeb *CC-MAIN-2024-10* snapshot for their educational quality on a scale from 0 to 5. We explored several prompt formats to automatically extract an educational score using an LLM and found that the additive scale used in previous work Yuan et al. [63] worked best. It allows the LLM to evaluate each criterion and build the score step-by-step, unlike the single-rating scale [64] which assigns a fixed score based on predefined categories. To avoid having the LLM favor highly technical pages like arXiv abstracts and submissions, we prompted it to focus on grade-school and middle-school level knowledge. The prompt used for synthetic annotations is in Appendix F.1.

To scale our filtering to the entirety of FineWeb, we trained a linear regression model on top of the *Snowflake-arctic-embed-m* embedding model [65]. We fine-tuned this linear regressor on 410,000 of our Llama 3 synthetic annotations for 20 epochs with a learning rate of 3e-4 (while keeping the embedding and encoder layers frozen). We selected the checkpoint with the highest F1 score on the held-out validation set containing the remaining 50,000 samples, treating Llama 3 annotations as ground-truth. After training, we rounded the model's output scores to integers from 0 to 5. We then used fixed thresholds to classify whether a given document from FineWeb was educational. We investigated the impact of using different thresholds for the filtering and ultimately chose a minimum threshold of 3 for FineWeb-Edu, which ultimately gave the best trade-off between performance on knowledge and reasoning intensive benchmarks and the performance on other benchmarks like HellaSwag [66]. With a threshold of 3, the model achieved an F1 score of 82% on the validation set, indicating strong performance in distinguishing high-quality educational content.

Applying the classifier to the 15 trillion tokens of FineWeb required 6,000 H100 GPU hours.

To confirm the effectiveness of education filtering at a larger scale, we conducted a larger ablation training a 1.71B model on 350 billion tokens, similar to the FineWeb filtering ablations mentioned

above. As shown in Fig. 10 and Fig. 11, we observed that FineWeb-Edu surpasses FineWeb and all other open web datasets, with quite remarkable improvements on educational benchmarks such as MMLU, ARC and OpenBookQA. Specifically, MMLU score increases from 33% to 37%, a relative improvement of approximately 12%, and ARC score goes from 46% to 57%, an improvement of about 24%.

On MMLU, FineWeb-Edu can match the final performance of Matrix with almost 10x fewer tokens, demonstrating the effectiveness of classifiers trained on LLM annotations for large-scale data filtering. Additional evaluation plots can be found in Appendix F.2.

## 4.1 Topic distribution

To examine how the educational classifier may skew the dataset towards certain topics, we embed [67] 50k samples from FineWeb and 50k samples from FineWeb-Edu using a sentence-transformers [68] model (all-MiniLM-L6-v2) which we then project to 2D space using UMAP [69]. Finally, we use DBSCAN [70] clustering to find the 100 densest topic clusters in the union of the two datasets, which we label using Llama 3.1 70B [6]. To compare the two datasets, we plot the difference of the size (as a percentage of the entire dataset) of each cluster in FineWeb-Edu and FineWeb in Fig. 18. The educational classifier heavily favors topics such as 'Education, Learning, Teaching' or 'History, Culture, Politics', while down-sampling 'Business, Finance, Law', 'Entertainment, Film, Theater' and 'Places, Travel, Real Estate', among others.

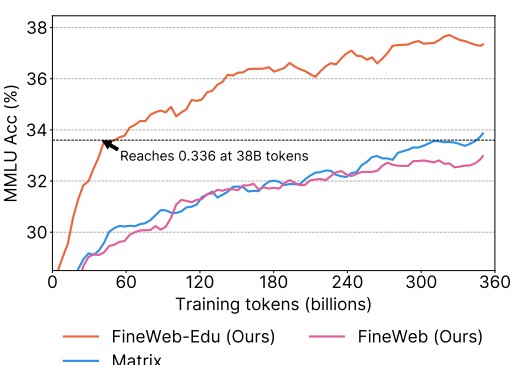

Figure 11: **Performance Comparison on MMLU**. FineWeb-Edu achieves a 33.6% accuracy on the MMLU benchmark at only 38 billion tokens, significantly outperforming Matrix (second best on the metric), which reaches similar accuracy at 300 billion tokens.

## 4.2 Domain fit

We evaluate the macro average perplexity of six checkpoints from our FineWeb and FineWeb-Edu ablation models on the domains from Paloma [71]. We use the codebase provided in [71] but intentionally do not perform decontamination, to compare how well each dataset covers different domains. The results are in Fig. 12. The FineWeb model generally shows lower perplexity in broad web sources such as C4, mC4, Falcon, Dolma V1.5 or the CommonCrawl subset of RedPajama, as well as on Twitter AAE, Manosphere, Gab, reddit (100 Subreddits) or 4chan. FineWeb-Edu tends to favour sources containing Wikipedia (WikiText-103 and M2D2 Wikipedia) or that are heavy in academic content (M2D2 S2ORC, which has semantic scholar papers, and the Arxiv subset of RedPajama). FineWeb-Edu also seems to have better coverage of programming content (100 PLs) than FineWeb. We have included results per subset for some of these sources in Appendix F.4.

## 5 Bias analyses

Language models are known to reflect the biases present in their pretraining datsts [72–77]. To provide a brief picture of dataset bias in FineWeb and FineWeb-Edu, we focus on subgroups recognised as "sensitive" or "protected" in English-speaking countries. These are a subset of subgroups that are historically subject to discrimination and are disproportionately the target of negative societal norms such as stereotyping, which is reflected in text-based language consumed for a dataset. We find that the FineWeb dataset has a relative overrepresentation of words that reflect hegemonic norms, known to be overrepresented in online text [72], such as 'man' and 'christian'. Although biases across the *gender*, *religion*, and *age* subgroups examined are not strong, we see the most skewed association between religion words and intimacy, such as 'christian dating' and 'jewish singles'. Fittingly, the FineWeb-Edu dataset captures associations that are less tied to intimacy compared to FineWeb and more expected from educational content of history and health, such as 'man' being associated to 'king', and 'woman' associated to 'pregnancy'. Further details are provided in Appendix G.

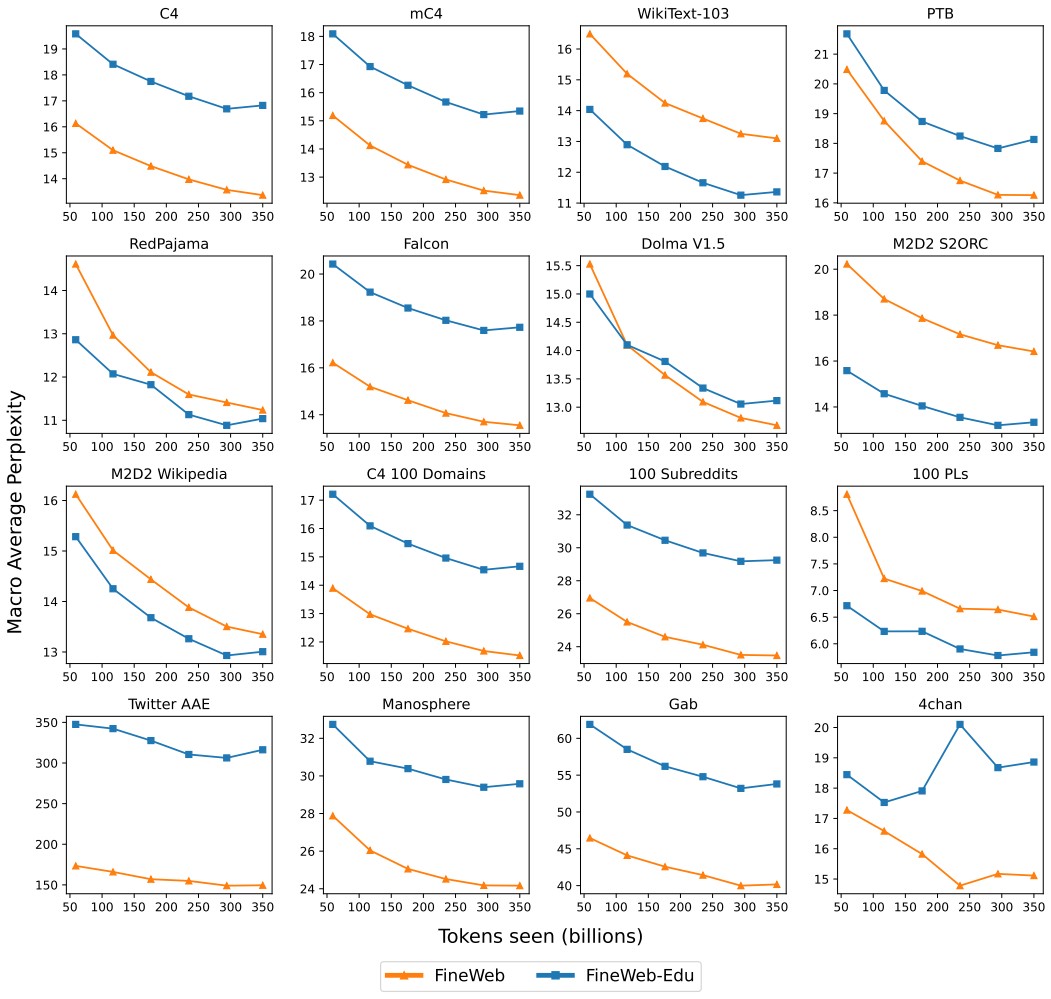

Figure 12: **FineWeb and FineWeb-Edu fit to Paloma domains**. FineWeb has lower perplexity on broad web sources while FineWeb-Edu has better coverage of Wikipedia and programming content.

# 6 Conclusion

In this paper, we developed the FineWeb datasets, a collection of large-scale LLM pretraining datasets that produce performant LLMs. Specifically, we release FineWeb, a 15-trillion token dataset derived from 96 Common Crawl snapshots, as well as FineWeb-Edu, a 1.3-trillion token dataset of educational content from FineWeb. FineWeb was created through a series of experiments that provided empirical evidence for our choice of text extraction strategy, deduplication procedure, and content filters. Both datasets are publicly released, along with the code and processing library that we used and all of the models we trained during our dataset ablation experiments.

While FineWeb and FineWeb-Edu attain state-of-the-art performance among public LLM pretraining datasets, we identify various paths to further improvement. First, both datasets are entirely comprised of web content scraped by Common Crawl. It is possible that augmenting either datasets with other datatypes (books, speech transcripts, etc.) could further improve performance. In addition, most of the experiments we ran were at a smaller scale due to computational constraints. Designing datasets at more realistic scales could provide more reliable guidance. Our evaluation setup was also by necessity limited to performance on academic benchmarks without any further instruction tuning or alignment. An evaluation setup that better reflected current usage patterns of LLMs might also be more reliable. We hope that our released datasets, code, and models help further improve public knowledge and development of performant LLM pretraining datasets.

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

## A FineWeb Datasheet

| Dataset Details | |
|---|---|
| Purpose of the dataset | We released FineWeb to make large language model training more accessible to the machine learning community at large. |
| Curated by | The dataset was curated by Hugging Face. |
| Funded by | The dataset was funded by Hugging Face. |
| Language(s) | English |
| License | The dataset is released under the Open Data Commons Attribution License (ODC-By) v1.0 license. The use of this dataset is also subject to CommonCrawl's Terms of Use. |
| **Dataset Structure** | |
| Data Instances | The following is an example sample from the dataset. It is part of the `CC-MAIN-2021-43` snapshot and was crawled on `2021-10-15T21:20:12Z`: |

```
{
  "text": "This is basically a
  ↪   peanut flavoured cream
  ↪   thickened with egg yolks and
  ↪   then set into a ramekin on top
  ↪   of some jam. Tony, one of the
  ↪   Wedgwood chefs, suggested
  ↪   sprinkling on some toasted
  ↪   crushed peanuts at the end to
  ↪   create extra crunch, which I
  ↪   thought was a great idea. The
  ↪   result is excellent.",
  "id":
  ↪   "<urn:uuid:e5a3e79a-13d4-4147-
  ↪   a26e-167536fcac5d>",
  "dump": "CC-MAIN-2021-43",
  "url": "<http://allrecipes.co.uk
  ↪   /recipe/24758/peanut-butter-and
  ↪   -jam-creme-brulee.aspx
  ↪   ?o_is=SimilarRecipes&o_ln=Sim
  ↪   Recipes_Photo_7>",
  "date": "2021-10-15T21:20:12Z",
  "file_path":
  ↪   "s3://commoncrawl/crawl-data/
  ↪   CC-MAIN-2021-43/segments/
  ↪   1634323583083.92/warc/
  ↪   CC-MAIN-20211015192439
  ↪   -20211015222439-00600.warc.gz",
  "language": "en",
  "language_score": 0.948729,
  "token_count": 69
}
```

| | |
|---|---|
| Data Fields | - `text` (`string`): the main text content
- `id` (`string`): original unique identifier for this sample from CommonCrawl
- `dump` (`string`): the CommonCrawl dump/snapshot this sample was a part of
- `url` (`string`): url to the original page where text was present
- `date` (`string`): crawl date (from CommonCrawl)
- `file_path` (`string`): s3 path for the individual CommonCrawl warc file containing this sample
- `language` (`string`): en for all the samples in this dataset
- `language_score` (`float`): language prediction score (0.01.0) as reported by the fastText language classifier
- `token_count` (`int`): number of tokens when applying the gpt2 tokenizer to this sample |
| Data Splits | The default subset includes the entire dataset. We also include separate splits for each CommonCrawl dump. FineWeb-Edu, a subset filtered for educational content, is also available. |
| **Dataset Creation** | |
| Curation Rationale | With FineWeb, we aim to provide the open source community with a clean and large-scale dataset for pretraining performant large language models. |
| Source Data | The source data consists of webpages crawled by the CommonCrawl foundation over the 2013-2024 time period. We then extracted the main page text from the HTML of each webpage, filtered each sample and deduplicated each individual CommonCrawl dump/crawl. |
| Data processing steps | The data processing pipeline consists of:

```- URL filtering```
```- Trafilatura text extraction```
```- FastText language filter```
```- MassiveText repetition and quality```
```↪  filters```
```- C4 quality filters```
```- FineWeb custom filters```
```- MinHash deduplication```
```- PII reformatting```

For FineWeb-Edu, we further apply a filtering step based on our educational content classifier. |
| Annotations | We augment the original samples with the `language`, `language_score` and `tokens_count` annotations. The language related annotations are automatically generated by our language filter. `token_count` is generated by applying the GPT-2 tokenizer to the text column. |
| Personal and Sensitive Information | We anonymize email addresses and public IP addresses using regex patterns. |

| Considerations for Using the Data | |
|---|---|
| Social Impact of Dataset | With the release of FineWeb, we aim to make LLM training more accessible to the machine learning community by:
(a) making the dataset creation process more transparent, by sharing our entire processing setup including the codebase used
(b) helping alleviate the costs of dataset curation, both in time and in compute, for model creators by publicly releasing our dataset with the community. |
| Biases | Efforts were made to minimize the amount of NSFW and toxic content present in the dataset by employing filtering on the URL level. However, there are still a significant number of documents present in the final dataset that could be considered to be toxic or contain harmful content. As FineWeb was sourced from the web as a whole, any harmful biases typically present in the web may be reproduced on our dataset. Bias analyses for sensitive subgroups demonstrate that 'man' is more common in the dataset than other gender terms, 'christian' is more common than other religion terms. The disproportionate association of specific terms to sensitive subgroups is relatively low, with the most notable bias that some religion terms tend to be more associated with online dating terms. We provide a more detailed bias analysis in Section 5. |
| Other Known Limitations | As a consequence of some of the filtering steps applied, it is likely that code content is not prevalent in our dataset. Users are advised to consider complementing FineWeb with other code datasets and specialized curated sources, such as Wikipedia, which may have better formatting than the Wikipedia content included in FineWeb. |

## B  License and hosting

The FineWeb datasets are released under the Open Data Commons Attribution License (ODC-By) v1.0. The full text of the license is available at https://opendatacommons.org/licenses/by/1-0/. The use of the dataset is also subject to CommonCrawl's Terms of Use. The authors of this work are solely responsible for the content and the views presented herein. NeurIPS is not associated and shall bear no responsibility for the work presented, including the dataset itself.

The FineWeb datasets are hosted on the HuggingFace hub, where they will remain available for the foreseeable future. We plan to regularly update the dataset with new CommonCrawl snapshots as they are released.

# C   Linked resources

| Resource | URL |
|---|---|
| FineWeb repository (DOI 10.57967/hf/2493) | https://hf.co/datasets/HuggingFaceFW/fineweb |
| FineWeb Croissant metadata | https://hf.co/api/datasets/HuggingFaceFW/fineweb/croissant |
| FineWeb-Edu repository (DOI 10.57967/hf/2497) | https://hf.co/datasets/HuggingFaceFW/fineweb-edu |
| FineWeb-Edu Croissant metadata | https://hf.co/api/datasets/HuggingFaceFW/fineweb-edu/croissant |
| FineWeb Llama3 annotations | https://huggingface.co/datasets/HuggingFaceFW/fineweb-edu-llama3-annotations |
| Educational classifier | https://huggingface.co/HuggingFaceFW/fineweb-edu-classifier |
| Dataset comparison models | https://hf.co/collections/HuggingFaceFW/comparison-models-662457b0d213e8c14fe47f32 |
| Ablation models | https://hf.co/collections/HuggingFaceFW/data-experiments-665ed849020d8b66a5d9896f |
| Datatrove processing code to reproduce FineWeb | https://github.com/huggingface/datatrove/blob/main/examples/fineweb.py |
| Evaluation setup | https://hf.co/datasets/HuggingFaceFW/fineweb/blob/main/lighteval_tasks.py |

# D   Data ablation setup

## D.1   Model architecture

| Parameter | Value |
|---|---|
| Architecture | Llama |
| Number of attention heads | 32 |
| Number of hidden layers | 24 |
| Number of key-value heads | 32 |
| RMS Norm epsilon | 1e-05 |
| Tied word embeddings | True |
| Embedding size | 50257 |
| Total number of parameters | 1.71B |
| Random initialization std | 0.02 |
| Tokenizer | GPT2 |

## D.2 Distributed training setup

| Parameter | Value |
|---|---|
| Data parallelism (dp) | 64 |
| Tensor parallelism (tp) | 1 |
| Pipeline parallelism (pp) | 1 |
| Micro-batch size | 4 |
| Sequence length | 2048 |
| Batch accumulation per replica | 4 |

## D.3 Optimizer Configuration

| Parameter | Value |
|---|---|
| Adam beta1 | 0.9 |
| Adam beta2 | 0.95 |
| Adam epsilon | 1.0e-8 |
| Gradient clipping | 1.0 |
| Weight decay | 0.1 |
| Learning rate | 3e-4 |
| Warmup steps | 500 |
| Warmup style | linear |
| Decay style | cosine |
| Minimum decay LR | 3.0e-5 |

# E Deduplication

## E.1 Deduplication parameters

As mentioned in Section 3.4, we use 5-grams and 112 hash functions for our MinHash deduplication. Each 5-gram is hashed with each of the 112 hash functions, and a document signature is obtained by taking the minimum hash value (minhash) across all 5-grams for each hash function. We further split the resulting 112 minhashes into 14 buckets of 8 hashes each. Documents are matched if they have the same 8 minhashes in at least one of the 14 buckets.

With these parameters, the probability that two documents with a n-gram similarity ($s$) of 0.7, 0.75, 0.8 and 0.85 would be identified as duplicates would be 56%, 77%, 92% and 98.8%, respectively. This split therefore will match documents that are at least 75% similar with a high probability, and almost guarantee that documents with similarities of 85% or above will be matched. These values can be computed by taking the following probabilities: that the two documents would have the same value for a given hash function, $s$; that they do not have the same 8 minhashes in one bucket, $1 - s^8$; that they do not have the same 8 minhashes in any of the 14 buckets, $(1 - s^8)^{14}$; and finally that they have the same 8 minhashes on at least one of the 14 buckets, $1 - (1 - s^8)^{14}$.

See Fig. 13 for a match probability comparison between our setup with 112 hashes and the one from RefinedWeb, with 9000 hashes, divided into 450 buckets of 20 hashes.

While the high number of hash functions in RefinedWeb allows for a steeper, more well-defined cut off (document pairs with similarity near the threshold are more likely to be correctly identified), this

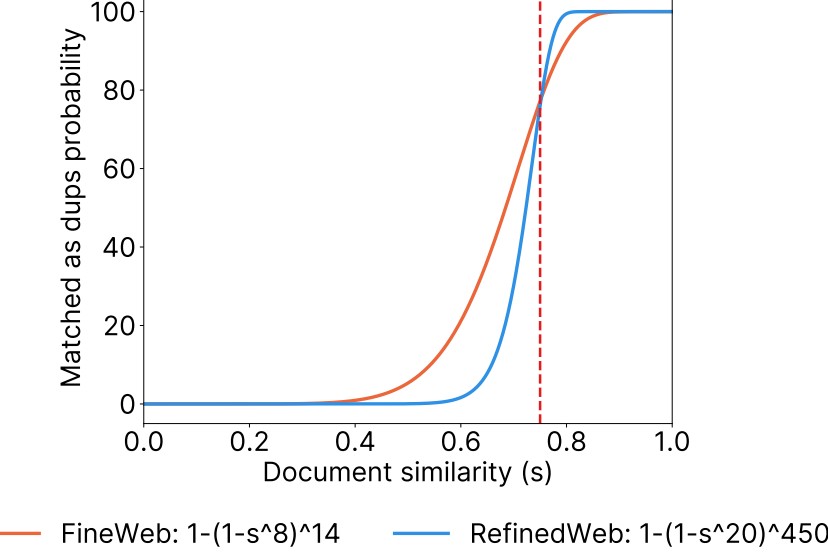

Figure 13: Comparison between FineWeb and RefinedWeb document matching probabilities.

larger number of hash functions also requires a substantially larger amount of compute resources, as each individual hash must be computed, stored, and then compared with hashes from other documents. We believe the compute and storage savings make up for the higher uncertainty on documents near the threshold.

## E.2 Measuring the effect of deduplication

Given the nature of deduplication, its effect is not always visible in a smaller slice of the dataset (such as 28B tokens, the size used for our filtering ablations). Furthermore, there are specific effects at play when deduplicating across different Common Crawl dumps, as some URLs and webpages are recrawled from one snapshot to the next.

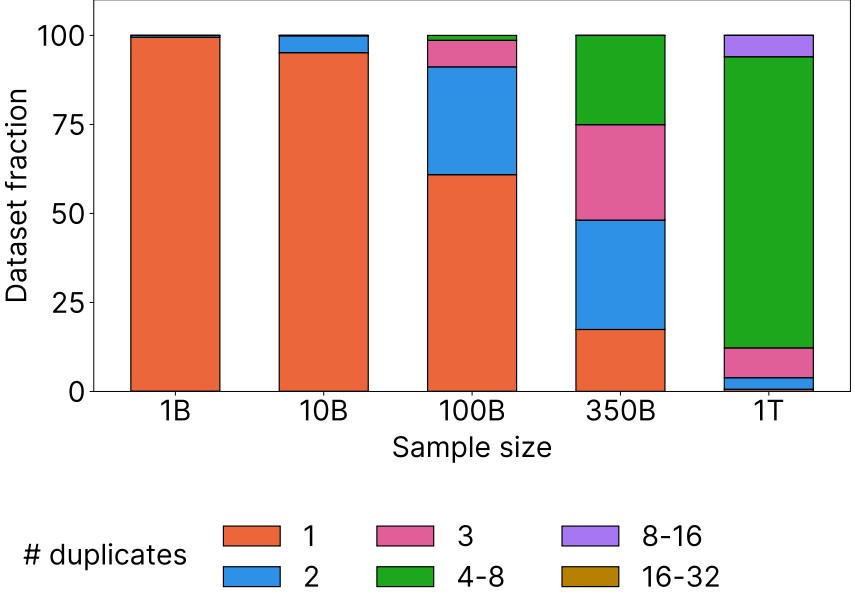

Figure 14: **Small ablations are ineffective for deduplication analysis.** The chart displays the distribution of document repetitions across different sample sizes (1 billion, 10 billion, 100 billion, 350 billion, and 1 trillion tokens) from a dataset of 20T tokens.

To visualize the effect of scaling the number of training tokens when measuring deduplication impact, we simulated creating different-sized subsets of randomly sampled documents from the full dataset under the following extreme conditions: there are 100 snapshots, where each one is made up of unique documents with a total of 200 billion tokens (yielding our total of 20 trillion from Section 3.4), and each snapshot is an exact copy of each other (worst case scenario for inter snapshot duplication).

In Fig. 14, we can see that for a 1 billion subset, almost all documents would be unique (#duplicates=1), despite each document being repeated 100 times in the full dataset. At the 100 billion scale (0.5% of the total dataset), there starts to be a larger number of documents being repeated twice, and a few even 4-8 times. At the larger scale of 1 trillion (5% of the total dataset), the majority of the documents are repeated up to 8 times, with some being repeated up to 16 times. This simulation illustrates the inherent difficulties with measuring deduplication impact on the training of larger LLMs once the largest duplicate clusters have been removed. We ran our performance evaluations for deduplicated data at the 350 billion scale, which would, under this theoretical scenario, be made up of a significant portion of documents duplicated up to 8 times.

## E.3   Alternative global deduplication

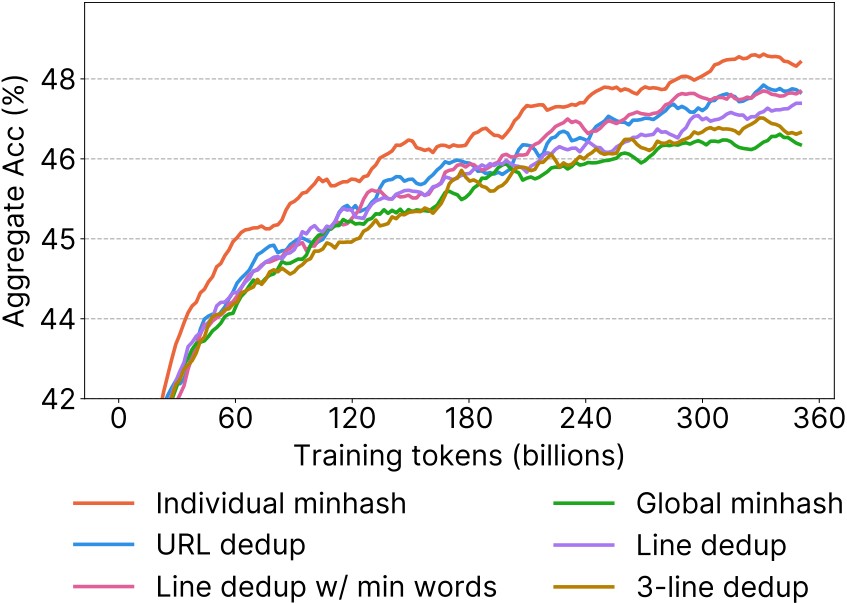

Figure 15: **URL and Line-wise deduplication study**. None of the attempted deduplication methods outperform individual deduplication.

To attempt to improve performance on top of independently deduplicating each snapshot, we experimented with applying other "lighter" global deduplication methods to all the individually MinHash deduplicated snapshots (comprising 20 trillion tokens of data).

We explored URL deduplication, where we only kept one document per normalized (lowercased) URL (71.5% of tokens removed, 5.6 trillion left) — *FineWeb URL dedup*. Different line-based deduplication variations were also considered: remove all but 1 (randomly chosen) occurrence of each duplicated line (77.8% of tokens dropped, 4.4 trillion left) — *FineWeb line dedup*; same as above, but only removing duplicate lines with at least 10 words and dropping documents with fewer than 3 sentences after deduplication (85% of tokens dropped, 2.9 trillion left) — *FineWeb line dedup w/ min words*; and remove all but 1 occurrence of each span of 3 duplicated lines with each number treated as 0 when finding duplicates, (80.9% of tokens removed, 3.7 trillion left) — *FineWeb 3-line dedup*.

As can be seen in Fig. 15 the performance of the models trained on each of these methods was consistently worse (albeit to different degrees) than that of the original individually deduplicated data. We therefore did not apply any additional deduplication beyond individual-snapshot MinHash-based deduplication.

### E.4 Other filters considered

| Metric | Threshold | Aggregate Acc (%) | Tokens removed (%) |
|---|---|---|---|
| lines-with-punct-ratio | $\geq 0.12$ | 42.85 | 10.14 |
| duplicated-line-char-ratio | $\leq 0.01$ | 42.78 | 12.47 |
| lines-with-punct-ratio | $\geq 0.12$ or $= 0$ | 42.72 | 5.82 |
| lines-shorter-30-ratio | $\leq 0.67$ | 42.65 | 3.37 |
| line-with-most-3-words-ratio | $\leq 0.49$ | 42.61 | 2.51 |
| duplicate-(5-10)-grams-char-ratio | $\leq 0.1, 0.084, 0.073, 0.065, 0.057, 0.05$ | 42.60 | 10.92 |
| lines-with-punct-ratio | $\geq 0.08$ or $= 0$ | 42.59 | 3.42 |
| top-(2,3,4)-gram-char-ratio | $\leq 0.13, 0.087, 0.079$ | 42.58 | 56.71 |
| lines-shorter-30-ratio | 0.69 | 42.58 | 3.73 |
| avg-words-per-line | $\geq 7$ | 42.56 | 2.32 |
| lines-shorter-30-ratio | $\leq 0.5$ | 42.53 | 11.17 |
| avg-words-per-line | $\geq 5$ | 42.39 | 0.83 |
| avg-words-per-line | $\geq 9$ | 42.27 | 4.47 |
| avg-line-length-0.5-sampling | $\geq 56$ | 42.93 | 3.24 |
| avg-line-length | $\geq 56$ | 42.12 | 6.48 |
| avg-line-length-0.5-sampling | $\geq 40$ | 42.03 | 1.50 |

Table 2: Full list of heuristic filters tested

# F    FineWeb-Edu

## F.1    Annotation Prompt

We use the following prompt template to generate document annotations using the Llama3 model:

> Below is an extract from a web page. Evaluate whether the page has a high educational value and could be useful in an educational setting for teaching from primary school to grade school levels using the additive 5-point scoring system described below. Points are accumulated based on the satisfaction of each criterion:
>
> - Add 1 point if the extract provides some basic information relevant to educational topics, even if it includes some irrelevant or non-academic content like advertisements and promotional material.
>
> - Add another point if the extract addresses certain elements pertinent to education but does not align closely with educational standards. It might mix educational content with non-educational material, offering a superficial overview of potentially useful topics, or presenting information in a disorganized manner and incoherent writing style.
>
> - Award a third point if the extract is appropriate for educational use and introduces key concepts relevant to school curricula. It is coherent though it may not be comprehensive or could include some extraneous information. It may resemble an introductory section of a textbook or a basic tutorial that is suitable for learning but has notable limitations like treating concepts that are too complex for grade school students.
>
> - Grant a fourth point if the extract highly relevant and beneficial for educational purposes for a level not higher than grade school, exhibiting a clear and consistent writing style. It could be similar to a chapter from a textbook or a tutorial, offering substantial educational content, including exercises and solutions, with minimal irrelevant information, and the concepts aren't too advanced for grade school students. The content is coherent, focused, and valuable for structured learning.
>
> - Bestow a fifth point if the extract is outstanding in its educational value, perfectly suited for teaching either at primary school or grade school. It follows detailed reasoning, the writing style is easy to follow and offers profound and thorough insights into the subject matter, devoid of any non-educational or complex content.
>
> The extract: <EXAMPLE>.
>
> After examining the extract:
>
> - Briefly justify your total score, up to 100 words.
>
> - Conclude with the score using the format: "Educational score: <total points>"

## F.2    Additional results

Fig. 16 compares FineWeb-Edu to other open web datasets on 9 becnhmarks, using a 1.71B model trained on 350 billion tokens. Additionally, Fig. 17 displays the results of experiments with various filtering thresholds for building FineWeb-Edu, using a 1.71B model trained on 28 billion tokens. Our findings indicate that a threshold of 3 yields the best average performance.

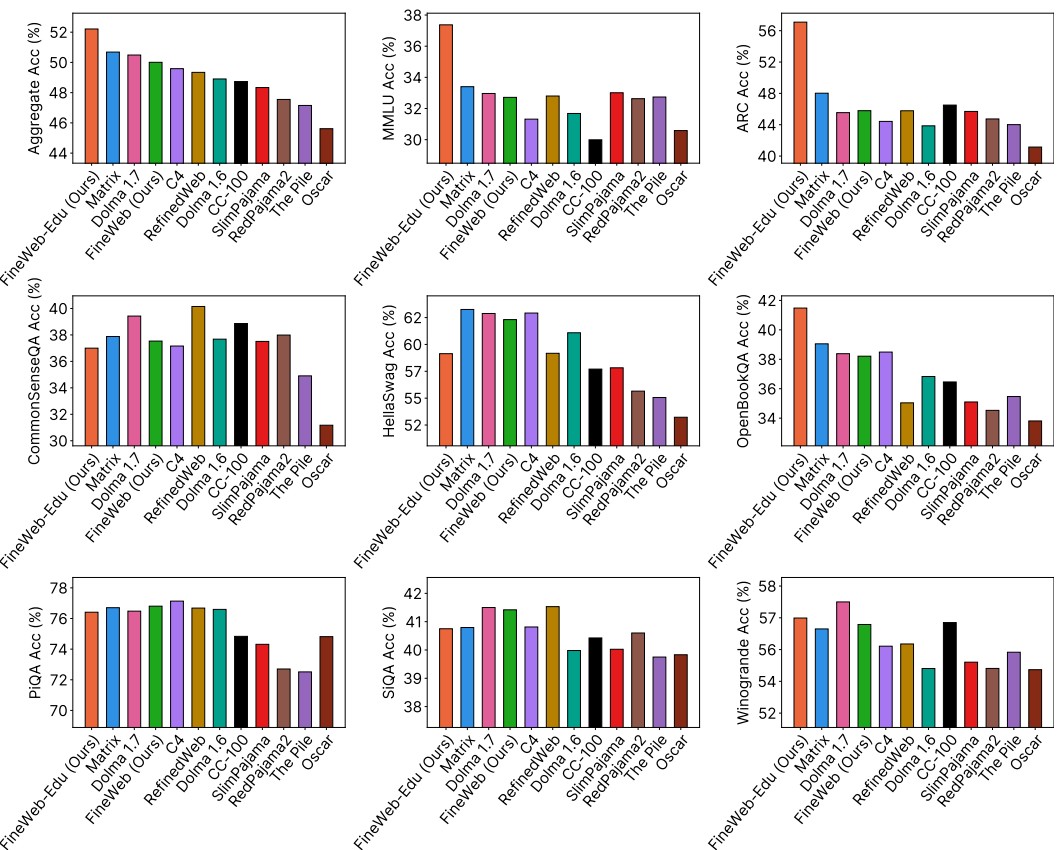

Figure 16: **Comparing FineWeb datasets to other public datasets on each benchmark.**

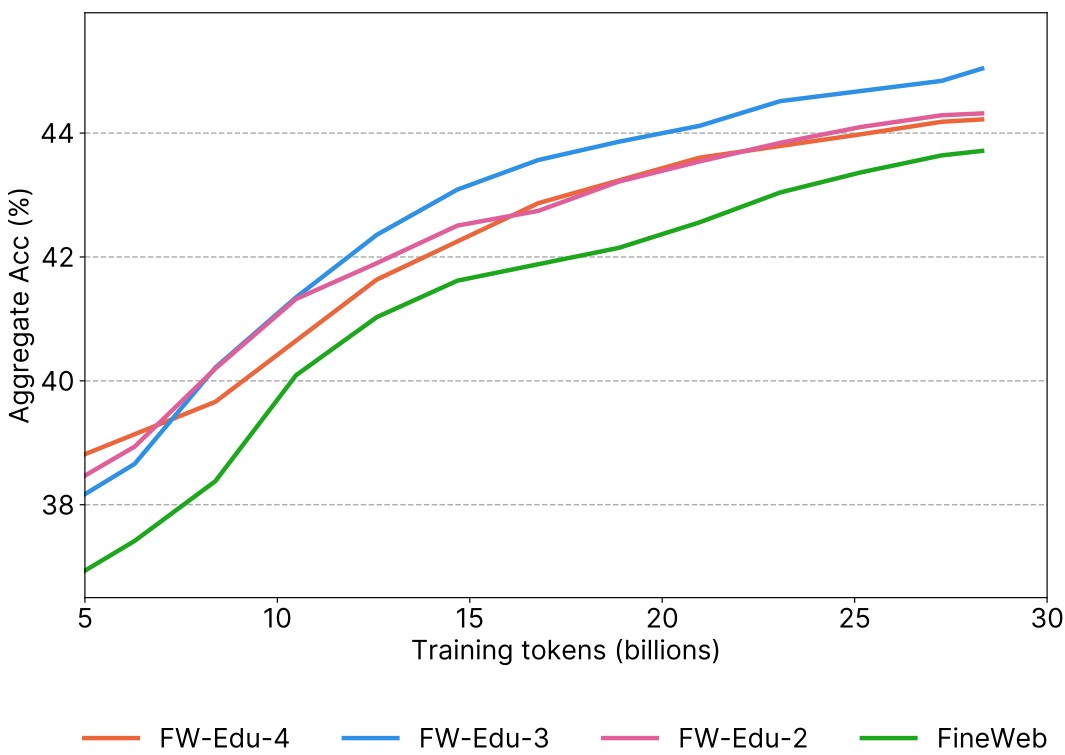

Figure 17: **Ablation study of FineWeb Edu thresholds**. Using a filtering threshold of 3 yields the best Aggregate Accuracy when building FineWeb-Edu. FW-Edu-$i$ denotes dataset filtered to only contain documents with an educational score greater or equal $i$.

## F.3 Topic distribution

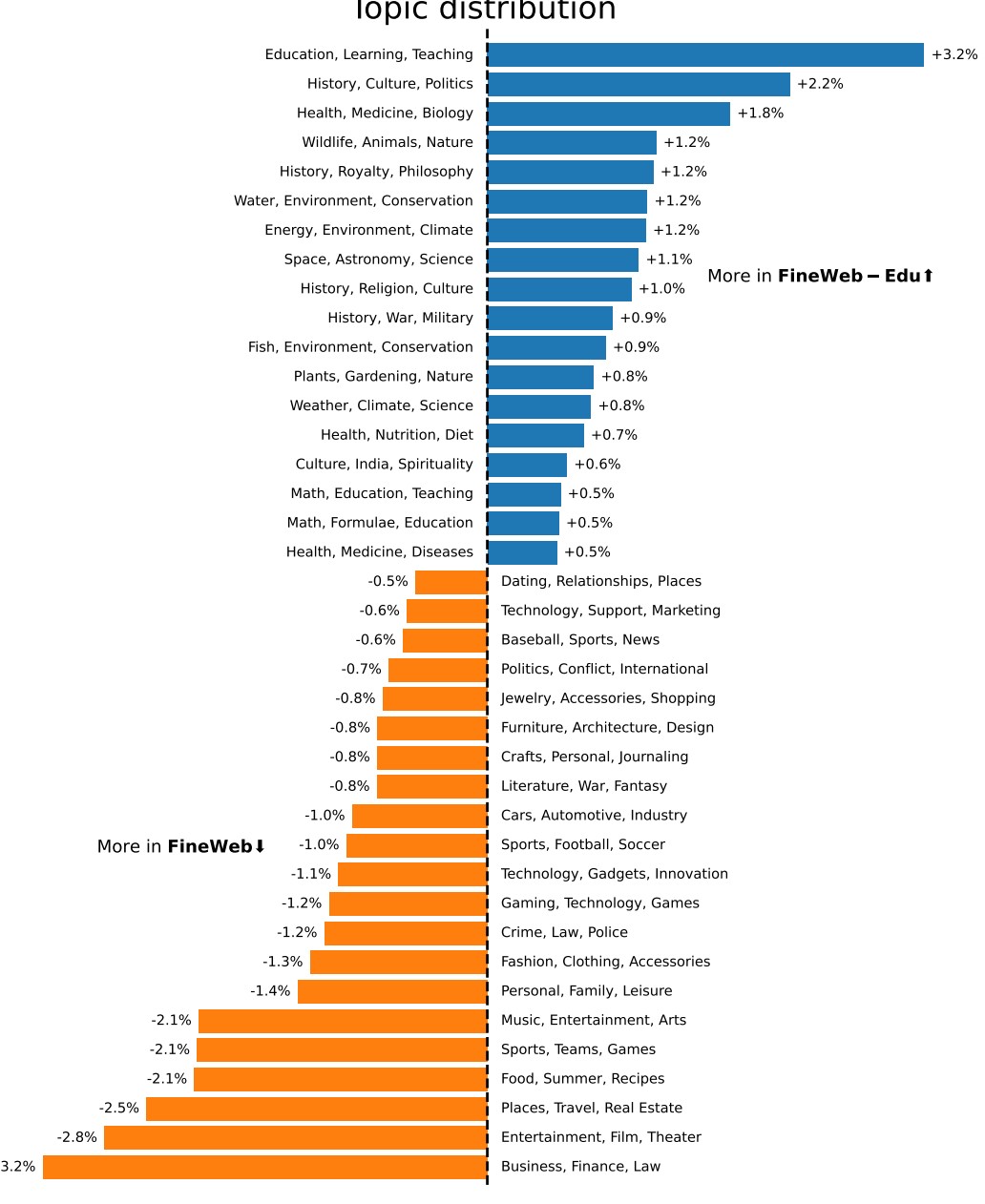

Figure 18: **FineWeb and FineWeb-Edu topic comparison**. FineWeb-Edu has a higher representation of topics like 'Education, Learning, Teaching' and 'History, Culture, Politics' compared to FineWeb. Conversely, it down-samples topics such as 'Business, Finance, Law' and 'Entertainment, Film, Theater.' Values indicate the absolute difference in the percentage of each topic between the datasets and only topics with an absolute difference of at least 0.5% are displayed.

## F.4 Domain fit

| Source | Domain | FineWeb ppl | FineWeb-Edu ppl |
|---|---|---|---|
| Dolma V1.5 | common-crawl | **14.499** | 18.336 |
| Dolma V1.5 | pes2o | 12.226 | **10.242** |
| Dolma V1.5 | reddit uniform | **23.814** | 29.864 |
| Dolma V1.5 | stack uniform | 7.65 | **7.014** |
| Dolma V1.5 | wiki | **12.0** | 12.243 |
| M2D2 Wikipedia | Culture and the arts | **10.367** | 14.518 |
| M2D2 Wikipedia | Culture and the arts Culture and Humanities | **14.037** | 14.116 |
| M2D2 Wikipedia | Culture and the arts Games and Toys | **15.774** | 18.912 |
| M2D2 Wikipedia | Culture and the arts Mass media | **14.352** | 18.134 |
| M2D2 Wikipedia | Culture and the arts Performing arts | 14.311 | **13.313** |
| M2D2 Wikipedia | Culture and the arts Sports and Recreation | **11.295** | 14.735 |
| M2D2 Wikipedia | Culture and the arts The arts and Entertainment | **13.669** | 19.039 |
| M2D2 Wikipedia | Culture and the arts Visual arts | **14.967** | 15.158 |
| M2D2 Wikipedia | General referece | 11.962 | **11.246** |
| M2D2 Wikipedia | General referece Further research tools and topics | **16.202** | 19.191 |
| M2D2 Wikipedia | General referece Reference works | **14.914** | 18.621 |
| M2D2 Wikipedia | Health and fitness | **12.0** | 13.448 |
| M2D2 Wikipedia | Health and fitness Exercise | **11.874** | 13.951 |
| M2D2 Wikipedia | Health and fitness Health science | 11.509 | **10.997** |
| M2D2 Wikipedia | Health and fitness Human medicine | **12.0** | 13.448 |
| M2D2 Wikipedia | Health and fitness Nutrition | 10.09 | **8.489** |
| M2D2 Wikipedia | Health and fitness Public health | 12.804 | **11.797** |
| M2D2 Wikipedia | Health and fitness Self care | 14.62 | **12.782** |
| M2D2 Wikipedia | History and events | 13.446 | **12.516** |
| M2D2 Wikipedia | History and events By continent | 14.174 | **12.066** |
| M2D2 Wikipedia | History and events By period | 12.94 | **11.0** |
| M2D2 Wikipedia | History and events By region | 13.61 | **11.63** |
| M2D2 Wikipedia | Human activites | **15.159** | 18.728 |
| M2D2 Wikipedia | Human activites Human activities | 12.784 | **11.117** |
| M2D2 Wikipedia | Human activites Impact of human activity | 15.092 | **13.592** |
| M2D2 Wikipedia | Mathematics and logic | 12.703 | **9.903** |
| M2D2 Wikipedia | Mathematics and logic Fields of mathematics | 12.703 | **9.903** |
| M2D2 Wikipedia | Mathematics and logic Logic | 14.281 | **13.367** |
| M2D2 Wikipedia | Mathematics and logic Mathematics | 14.923 | **14.207** |
| M2D2 Wikipedia | Natural and physical sciences | 12.884 | **10.529** |
| M2D2 Wikipedia | Natural and physical sciences Biology | 12.718 | **10.221** |
| M2D2 Wikipedia | Natural and physical sciences Earth sciences | 15.346 | **13.145** |

| Source | Domain | FineWeb ppl | FineWeb-Edu ppl |
|---|---|---|---|
| M2D2 Wikipedia | Natural and physical sciences Nature | 12.594 | **9.886** |
| M2D2 Wikipedia | Natural and physical sciences Physical sciences | 13.088 | **10.643** |
| M2D2 Wikipedia | Philosophy and thinking | **14.081** | 16.067 |
| M2D2 Wikipedia | Philosophy and thinking Philosophy | 14.209 | **12.91** |
| M2D2 Wikipedia | Philosophy and thinking Thinking | **14.081** | 16.067 |
| M2D2 Wikipedia | Religion and belief systems | 12.636 | **11.326** |
| M2D2 Wikipedia | Religion and belief systems Allah | 14.072 | **10.808** |
| M2D2 Wikipedia | Religion and belief systems Belief systems | 12.843 | **11.652** |
| M2D2 Wikipedia | Religion and belief systems Major beliefs of the world | 13.824 | **11.834** |
| M2D2 Wikipedia | Society and social sciences | 11.777 | **11.195** |
| M2D2 Wikipedia | Society and social sciences Social sciences | **11.81** | 13.03 |
| M2D2 Wikipedia | Society and social sciences Society | 11.777 | **11.195** |
| M2D2 Wikipedia | Technology and applied sciences | 11.592 | **9.368** |
| M2D2 Wikipedia | Technology and applied sciences Agriculture | **13.941** | 14.998 |
| M2D2 Wikipedia | Technology and applied sciences Computing | **15.562** | 16.091 |
| M2D2 Wikipedia | Technology and applied sciences Engineering | 14.897 | **13.861** |
| M2D2 Wikipedia | Technology and applied sciences Transport | **16.519** | 17.886 |
| Manosphere | avfm | **27.332** | 32.058 |
| Manosphere | incels | **18.253** | 20.788 |
| Manosphere | love shy | **28.206** | 33.374 |
| Manosphere | mgtow | **24.913** | 29.702 |
| Manosphere | pua forum | **25.133** | 33.297 |
| Manosphere | red pill talk | **33.87** | 42.947 |
| Manosphere | reddit | **24.786** | 30.903 |
| Manosphere | rooshv | **23.593** | 27.819 |
| Manosphere | the attraction | **24.988** | 30.907 |
| RedPajama | arxiv | 32.338 | **23.368** |
| RedPajama | books | **22.095** | 23.953 |
| RedPajama | c4 | **12.685** | 15.599 |
| RedPajama | commoncrawl | **8.0** | 8.979 |
| RedPajama | github | 5.613 | **5.247** |
| RedPajama | stackexchange | 9.055 | **8.862** |
| RedPajama | wikipedia | 8.741 | **8.608** |
| Twitter AAE | AA | **246.907** | 575.106 |
| Twitter AAE | white | **98.536** | 192.374 |

Table 3: **Paloma domain comparison between FineWeb and FineWeb-Edu**. Lower perplexity (ppl) in bold. A lower perplexity value indicates a better fit to a given domain.

## G    Bias Analyses

### G.1    Distributional Analysis

| Subgroup | Terms |
|----------|-------|
| *age* | 'old', 'young' |
| *gender* | 'man', 'woman', 'non-binary' |
| *religion* | 'muslim', 'christian', 'jewish', 'hindu', 'buddhist', 'atheist' |

Table 4: Subgroups and terms used for bias analyses.

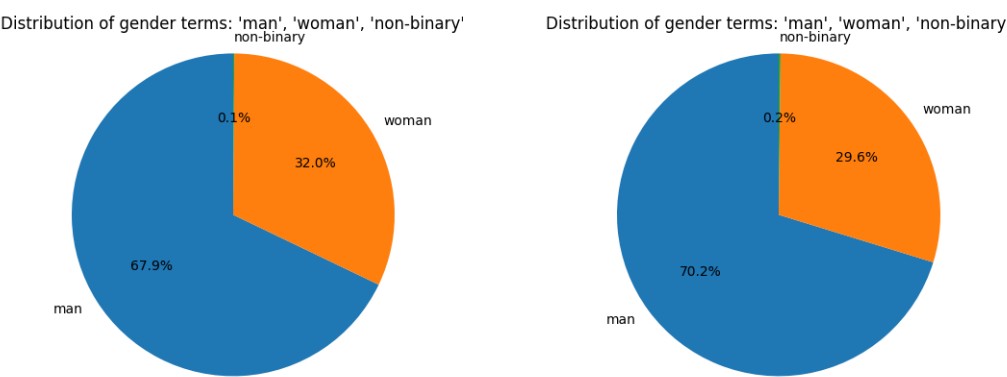

Figure 19: Distribution of *gender* terms in FineWeb (Left) and FineWeb-Edu (Right), 10BT samples.

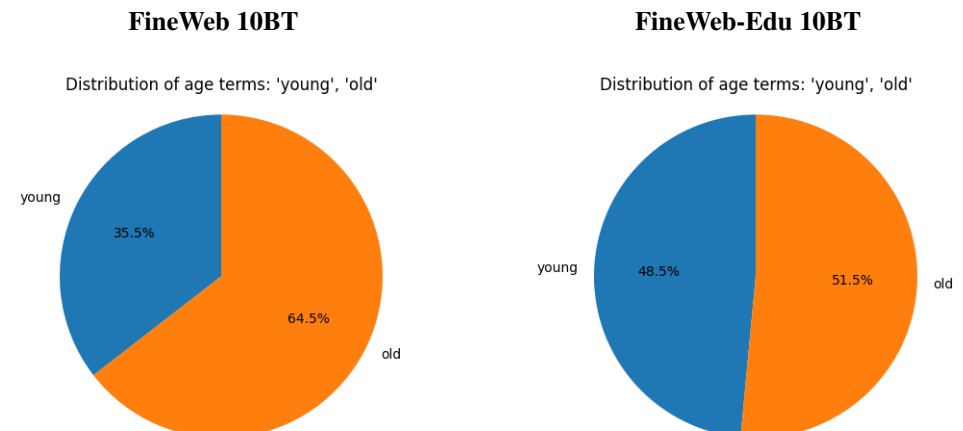

Figure 20: Distribution of *age* terms in FineWeb (Left) and FineWeb-Edu (Right), 10BT samples.

To begin, we examine the distribution over subgroup terms for *gender* (Fig. 19) *age* (Fig. 20), and *religion* (Fig. 21) in a subset of FineWeb and FineWeb-Edu randomly sampled from the whole dataset, of around 10 Billion GPT-2 tokens (FineWeb 10BT and FineWeb-Edu 10BT). Terms used are shown in Table 4 and are all normalized to lowercase for this analysis.

We find that 'man' appears much more frequently than 'woman' and 'non-binary', and 'christian' appears much more frequently than all other religions terms tested.

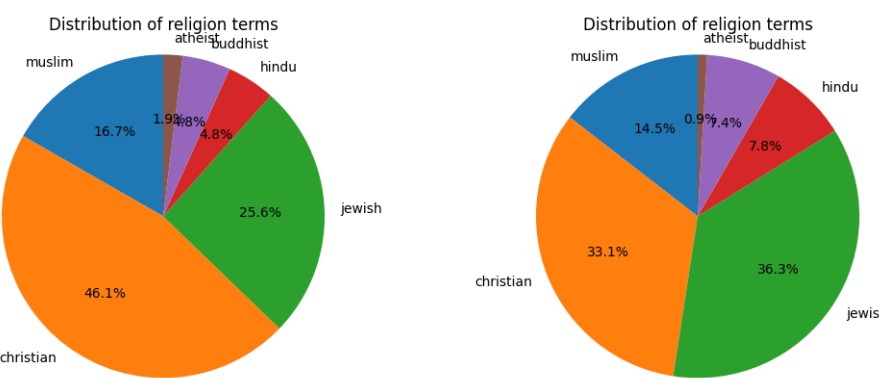

Figure 21: Distribution of *religion* terms in FineWeb (Left) and FineWeb-Edu (Right), 10BT samples.

### G.2  Association Analysis

We next examine the skews with respect to the different subgroup terms, as measured by TF-IDF [78]. This method is described as capturing the *specificity* of words in the dataset, here applied as specificity with respect to the terms for the different subgroups. This provides a way to quantify how "biased" each subgroup term is with respect to the words they co-occur with. Specifically, given the dataset and terms for a subgroup of interest, we:

1. Build a vocabulary of all words that occur at least twice in the dataset.
2. Extract all data instances where the subgroup term is present.
3. Compute the TF-IDF for all words in the vocabulary that co-occur in the same documents as a given subgroup term.
4. Compute the difference between the TF-IDF for the given subgroup terms and the average TF-IDF of all other words they co-occur with.
5. Extract the words co-occurring with the given subgroup terms with a TF-IDF greater than 0.

#### G.2.1  Gender

We find that 'man' is associated with terms such as 'god', 'police', 'said' and 'good', 'woman' is associated with terms like 'said', 'women', 'police', 'life', 'love', 'dating' and 'family', and 'non-binary' is associated with 'gender' and LGBTQIA+ terms such as 'trans', 'transgender', and 'queer' (Fig. 22). Applying this same analysis to FineWeb-Edu-Sample-10BT, we find that 'man' is associated with the term 'god', and slightly associated with terms like 'war', 'great', and 'king'. 'woman' is associated with terms like 'pregnancy', 'cancer', 'mother', 'children', and 'family'.

#### G.2.2  Religion

Throughout, we see skews towards words associated with online intimacy: 'online', 'singles', 'sex', 'mature', 'girls'. As can be seen in Fig. 27, 'jewish' is particularly associated with 'dating' and 'singles'. 'muslim', 'jewish', 'hindu' and 'buddhist' are slightly skewed to co-occur with 'women', while 'sex' is skewed with 'muslim', 'christian', 'jewish'; and 'girl' with 'muslim', 'jewish', 'hindu'.

#### G.2.3  Age

The word 'young' is skewed to co-occur with 'women', consistent with the problematic tendencies in English-speaking societies to infantilize women and over-indexing on womens' youth [79, 80]. We also see expected skews, such as 'young' co-occurring with words like 'children' and 'school'.

**A**

| word | non-binary | non-binary+ | man | man+ | woman | woman+ |
|---|---|---|---|---|---|---|
| non-binary | 0.092 | 0.061 | 0.000 | -0.031 | 0.000 | -0.031 |
| gender | 0.068 | 0.044 | 0.001 | -0.023 | 0.003 | -0.021 |
| trans | 0.055 | 0.037 | 0.000 | -0.018 | 0.001 | -0.018 |
| transgender | 0.035 | 0.023 | 0.000 | -0.012 | 0.001 | -0.011 |
| queer | 0.033 | 0.021 | 0.000 | -0.011 | 0.001 | -0.011 |
| people | 0.044 | 0.016 | 0.018 | -0.009 | 0.020 | -0.007 |
| women | 0.044 | 0.015 | 0.011 | -0.018 | 0.031 | 0.003 |
| lgbtq | 0.020 | 0.013 | 0.000 | -0.007 | 0.000 | -0.006 |
| community | 0.020 | 0.011 | 0.004 | -0.006 | 0.004 | -0.005 |
| sexual | 0.017 | 0.008 | 0.003 | -0.005 | 0.006 | -0.003 |
| female | 0.016 | 0.007 | 0.003 | -0.006 | 0.006 | -0.002 |
| sex | 0.019 | 0.006 | 0.007 | -0.006 | 0.012 | -0.001 |
| work | 0.016 | 0.004 | 0.009 | -0.003 | 0.010 | -0.002 |
| person | 0.014 | 0.004 | 0.007 | -0.003 | 0.008 | -0.001 |
| feel | 0.012 | 0.003 | 0.006 | -0.003 | 0.008 | -0.001 |
| dating | 0.015 | 0.002 | 0.009 | -0.004 | 0.015 | 0.002 |
| ve | 0.012 | 0.002 | 0.009 | -0.001 | 0.010 | -0.000 |
| new | 0.015 | 0.001 | 0.013 | -0.001 | 0.013 | -0.000 |
| like | 0.022 | 0.001 | 0.019 | -0.002 | 0.021 | 0.000 |
| men | 0.015 | 0.001 | 0.013 | -0.001 | 0.014 | -0.000 |
| want | 0.012 | 0.001 | 0.009 | -0.002 | 0.011 | 0.000 |
| world | 0.013 | 0.001 | 0.011 | -0.000 | 0.011 | -0.001 |
| really | 0.012 | 0.001 | 0.010 | -0.001 | 0.011 | 0.000 |
| year | 0.010 | 0.001 | 0.008 | -0.000 | 0.008 | -0.001 |
| young | 0.010 | 0.001 | 0.008 | -0.001 | 0.009 | 0.000 |

**B**

| word | woman | woman+ | non-binary | non-binary+ | man | man+ |
|---|---|---|---|---|---|---|
| woman | 0.051 | 0.026 | 0.011 | -0.013 | 0.011 | -0.013 |
| said | 0.022 | 0.004 | 0.011 | -0.007 | 0.022 | 0.004 |
| women | 0.031 | 0.003 | 0.044 | 0.015 | 0.011 | -0.018 |
| police | 0.012 | 0.003 | 0.003 | -0.007 | 0.014 | 0.004 |
| life | 0.017 | 0.002 | 0.011 | -0.003 | 0.015 | 0.001 |
| love | 0.015 | 0.002 | 0.012 | -0.001 | 0.012 | -0.001 |
| dating | 0.015 | 0.002 | 0.015 | 0.002 | 0.009 | -0.004 |
| family | 0.010 | 0.002 | 0.007 | -0.002 | 0.009 | -0.000 |
| did | 0.011 | 0.002 | 0.006 | -0.003 | 0.012 | 0.002 |
| just | 0.019 | 0.002 | 0.015 | -0.002 | 0.018 | 0.000 |
| know | 0.015 | 0.002 | 0.012 | -0.002 | 0.014 | 0.000 |
| time | 0.017 | 0.001 | 0.013 | -0.002 | 0.017 | 0.001 |
| day | 0.012 | 0.001 | 0.008 | -0.002 | 0.011 | 0.001 |
| good | 0.011 | 0.001 | 0.007 | -0.003 | 0.012 | 0.002 |
| story | 0.011 | 0.001 | 0.009 | -0.001 | 0.009 | -0.000 |
| going | 0.010 | 0.001 | 0.007 | -0.002 | 0.010 | 0.001 |
| say | 0.010 | 0.001 | 0.008 | -0.002 | 0.010 | 0.001 |
| god | 0.012 | 0.001 | 0.003 | -0.008 | 0.018 | 0.007 |
| years | 0.012 | 0.001 | 0.010 | -0.001 | 0.011 | 0.001 |
| don | 0.014 | 0.001 | 0.013 | 0.000 | 0.012 | -0.001 |
| book | 0.010 | 0.001 | 0.009 | -0.000 | 0.008 | -0.001 |
| right | 0.009 | 0.001 | 0.008 | -0.001 | 0.009 | 0.000 |

**C**

| word | man | man+ | woman | woman+ | non-binary | non-binary+ |
|---|---|---|---|---|---|---|
| man | 0.046 | 0.022 | 0.019 | -0.005 | 0.007 | -0.017 |
| god | 0.018 | 0.007 | 0.012 | 0.001 | 0.003 | -0.008 |
| police | 0.014 | 0.004 | 0.012 | 0.003 | 0.003 | -0.007 |
| said | 0.022 | 0.004 | 0.022 | 0.004 | 0.011 | -0.007 |
| good | 0.012 | 0.002 | 0.011 | 0.001 | 0.007 | -0.003 |
| did | 0.012 | 0.002 | 0.011 | 0.002 | 0.006 | -0.003 |
| say | 0.010 | 0.001 | 0.010 | 0.001 | 0.008 | -0.002 |
| time | 0.017 | 0.001 | 0.017 | 0.001 | 0.013 | -0.002 |
| day | 0.011 | 0.001 | 0.012 | 0.001 | 0.008 | -0.002 |
| going | 0.010 | 0.001 | 0.010 | 0.001 | 0.007 | -0.002 |
| years | 0.011 | 0.001 | 0.012 | 0.001 | 0.010 | -0.001 |
| life | 0.015 | 0.001 | 0.017 | 0.002 | 0.011 | -0.003 |

Figure 22: Most skewed associations in FineWeb for *gender* terms 'non-binary' (A), 'woman' (B), and 'man' (C) in FineWeb compared to one another, measured using TF-IDF. Columns are sorted by the 'non-binary+', 'woman+' and 'man+' columns, measuring the difference from the mean over all words occurring more than once in the dataset.

| word | atheist | atheist+ | muslim | muslim+ | christian | christian+ | jewish | jewish+ | hindu | hindu+ | buddhist | buddhist+ |
|---|---|---|---|---|---|---|---|---|---|---|---|---|
| atheist | 0.095 | 0.078 | 0.001 | -0.016 | 0.003 | -0.014 | 0.001 | -0.016 | 0.001 | -0.016 | 0.001 | -0.016 |
| god | 0.080 | 0.057 | 0.010 | -0.013 | 0.026 | 0.003 | 0.008 | -0.015 | 0.008 | -0.014 | 0.006 | -0.017 |
| religion | 0.043 | 0.030 | 0.010 | -0.004 | 0.007 | -0.007 | 0.005 | -0.009 | 0.010 | -0.004 | 0.008 | -0.006 |
| religious | 0.039 | 0.028 | 0.007 | -0.004 | 0.006 | -0.006 | 0.005 | -0.006 | 0.006 | -0.005 | 0.006 | -0.006 |
| church | 0.027 | 0.017 | 0.004 | -0.006 | 0.016 | 0.006 | 0.005 | -0.005 | 0.003 | -0.007 | 0.005 | -0.005 |
| people | 0.035 | 0.014 | 0.021 | -0.001 | 0.019 | -0.002 | 0.019 | -0.002 | 0.016 | -0.006 | 0.019 | -0.003 |
| think | 0.021 | 0.012 | 0.008 | -0.001 | 0.008 | -0.001 | 0.007 | -0.002 | 0.004 | -0.005 | 0.007 | -0.002 |
| don | 0.021 | 0.011 | 0.009 | -0.001 | 0.009 | -0.001 | 0.007 | -0.002 | 0.004 | -0.005 | 0.007 | -0.002 |
| life | 0.023 | 0.010 | 0.009 | -0.004 | 0.013 | -0.000 | 0.010 | -0.004 | 0.011 | -0.003 | 0.015 | 0.002 |
| like | 0.024 | 0.009 | 0.014 | -0.001 | 0.015 | -0.001 | 0.014 | -0.002 | 0.011 | -0.004 | 0.014 | -0.002 |
| know | 0.020 | 0.009 | 0.010 | -0.001 | 0.011 | 0.000 | 0.009 | -0.001 | 0.006 | -0.004 | 0.008 | -0.003 |
| just | 0.022 | 0.009 | 0.013 | -0.001 | 0.014 | -0.000 | 0.012 | -0.002 | 0.009 | -0.005 | 0.013 | -0.001 |
| world | 0.020 | 0.008 | 0.012 | -0.000 | 0.010 | -0.002 | 0.011 | -0.001 | 0.010 | -0.003 | 0.010 | -0.002 |
| way | 0.016 | 0.006 | 0.008 | -0.001 | 0.009 | -0.000 | 0.008 | -0.002 | 0.006 | -0.003 | 0.010 | 0.000 |
| good | 0.015 | 0.006 | 0.009 | -0.001 | 0.010 | 0.001 | 0.008 | -0.002 | 0.006 | -0.003 | 0.008 | -0.001 |
| time | 0.017 | 0.005 | 0.011 | -0.001 | 0.011 | -0.000 | 0.010 | -0.001 | 0.008 | -0.003 | 0.011 | -0.000 |
| man | 0.015 | 0.004 | 0.012 | 0.001 | 0.012 | 0.001 | 0.011 | 0.000 | 0.009 | -0.002 | 0.007 | -0.004 |
| christian | 0.031 | 0.003 | 0.016 | -0.012 | 0.077 | 0.049 | 0.018 | -0.009 | 0.013 | -0.015 | 0.011 | -0.016 |
| catholic | 0.012 | 0.002 | 0.006 | -0.004 | 0.015 | 0.005 | 0.014 | 0.003 | 0.006 | -0.004 | 0.010 | -0.001 |

Figure 23: Most skewed associations in FineWeb for 'atheist' compared to other religions, measured using TF-IDF. Columns are sorted by the 'atheist+' column, measuring the difference from the mean over all words.

| word | buddhist | buddhist+ | atheist | atheist+ | muslim | muslim+ | christian | christian+ | jewish | jewish+ | hindu | hindu+ |
|---|---|---|---|---|---|---|---|---|---|---|---|---|
| buddhist | 0.169 | 0.134 | 0.003 | -0.033 | 0.006 | -0.029 | 0.005 | -0.030 | 0.007 | -0.029 | 0.022 | -0.013 |
| single | 0.055 | 0.018 | 0.003 | -0.034 | 0.034 | -0.003 | 0.033 | -0.004 | 0.045 | 0.007 | 0.054 | 0.017 |
| singles | 0.085 | 0.015 | 0.002 | -0.068 | 0.065 | -0.005 | 0.076 | 0.006 | 0.110 | 0.041 | 0.079 | 0.010 |
| personals | 0.034 | 0.012 | 0.001 | -0.021 | 0.018 | -0.004 | 0.018 | -0.004 | 0.032 | 0.009 | 0.031 | 0.009 |
| site | 0.038 | 0.007 | 0.004 | -0.027 | 0.033 | 0.002 | 0.035 | 0.004 | 0.043 | 0.012 | 0.035 | 0.003 |
| men | 0.036 | 0.007 | 0.009 | -0.021 | 0.030 | 0.001 | 0.028 | -0.002 | 0.034 | 0.004 | 0.040 | 0.011 |
| women | 0.046 | 0.006 | 0.010 | -0.030 | 0.048 | 0.007 | 0.037 | -0.004 | 0.050 | 0.009 | 0.053 | 0.012 |
| chat | 0.019 | 0.005 | 0.001 | -0.013 | 0.014 | 0.000 | 0.016 | 0.002 | 0.016 | 0.002 | 0.017 | 0.003 |
| meet | 0.028 | 0.004 | 0.003 | -0.021 | 0.028 | 0.003 | 0.026 | 0.001 | 0.034 | 0.010 | 0.027 | 0.003 |
| 100 | 0.013 | 0.003 | 0.002 | -0.007 | 0.008 | -0.001 | 0.010 | 0.000 | 0.011 | 0.002 | 0.011 | 0.002 |
| essay | 0.013 | 0.003 | 0.003 | -0.007 | 0.007 | -0.003 | 0.009 | -0.001 | 0.010 | 0.001 | 0.017 | 0.007 |
| free | 0.035 | 0.002 | 0.008 | -0.024 | 0.034 | 0.002 | 0.038 | 0.005 | 0.042 | 0.009 | 0.038 | 0.005 |
| date | 0.011 | 0.002 | 0.002 | -0.008 | 0.010 | 0.000 | 0.012 | 0.002 | 0.012 | 0.002 | 0.012 | 0.002 |
| life | 0.015 | 0.002 | 0.023 | 0.010 | 0.009 | -0.004 | 0.013 | -0.000 | 0.010 | -0.004 | 0.011 | -0.003 |
| asian | 0.011 | 0.001 | 0.001 | -0.009 | 0.011 | 0.002 | 0.009 | -0.001 | 0.012 | 0.002 | 0.014 | 0.004 |
| looking | 0.014 | 0.001 | 0.004 | -0.009 | 0.015 | 0.002 | 0.015 | 0.001 | 0.018 | 0.005 | 0.015 | 0.001 |

Figure 24: Most skewed associations in FineWeb for 'buddhist' compared to other religions, measured using TF-IDF. Columns are sorted by the 'buddhist+' column, measuring the difference from the mean over all words.

| word | christian | christian+ | jewish | jewish+ | hindu | hindu+ | buddhist | buddhist+ | atheist | atheist+ | muslim | muslim+ |
|---|---|---|---|---|---|---|---|---|---|---|---|---|
| dating | 0.192 | 0.049 | 0.212 | 0.069 | 0.146 | 0.004 | 0.133 | -0.009 | 0.009 | -0.134 | 0.164 | 0.021 |
| christian | 0.077 | 0.049 | 0.018 | -0.009 | 0.013 | -0.015 | 0.011 | -0.016 | 0.031 | 0.003 | 0.016 | -0.012 |
| online | 0.047 | 0.010 | 0.057 | 0.020 | 0.038 | 0.001 | 0.032 | -0.005 | 0.004 | -0.033 | 0.043 | 0.006 |
| sites | 0.023 | 0.009 | 0.022 | 0.008 | 0.013 | -0.002 | 0.009 | -0.005 | 0.002 | -0.013 | 0.019 | 0.004 |
| singles | 0.076 | 0.006 | 0.110 | 0.041 | 0.079 | 0.010 | 0.085 | 0.015 | 0.002 | -0.068 | 0.065 | -0.005 |
| church | 0.016 | 0.006 | 0.005 | -0.005 | 0.003 | -0.007 | 0.005 | -0.005 | 0.027 | 0.017 | 0.004 | -0.006 |
| free | 0.038 | 0.005 | 0.042 | 0.009 | 0.038 | 0.005 | 0.035 | 0.002 | 0.008 | -0.024 | 0.034 | 0.002 |
| catholic | 0.015 | 0.005 | 0.014 | 0.003 | 0.006 | -0.004 | 0.010 | -0.001 | 0.012 | 0.002 | 0.006 | -0.004 |
| site | 0.035 | 0.004 | 0.043 | 0.012 | 0.035 | 0.003 | 0.038 | 0.007 | 0.004 | -0.027 | 0.033 | 0.002 |
| love | 0.017 | 0.003 | 0.014 | -0.000 | 0.016 | 0.001 | 0.014 | -0.001 | 0.013 | -0.002 | 0.013 | -0.001 |
| god | 0.026 | 0.003 | 0.008 | -0.015 | 0.008 | -0.014 | 0.006 | -0.017 | 0.080 | 0.057 | 0.010 | -0.013 |
| chat | 0.016 | 0.002 | 0.016 | 0.002 | 0.017 | 0.003 | 0.019 | 0.005 | 0.001 | -0.013 | 0.014 | 0.000 |
| best | 0.015 | 0.002 | 0.016 | 0.003 | 0.014 | 0.001 | 0.013 | -0.000 | 0.006 | -0.006 | 0.014 | 0.001 |
| date | 0.012 | 0.002 | 0.012 | 0.002 | 0.012 | 0.002 | 0.011 | 0.002 | 0.002 | -0.008 | 0.010 | 0.000 |
| meet | 0.026 | 0.001 | 0.034 | 0.010 | 0.027 | 0.003 | 0.028 | 0.004 | 0.003 | -0.021 | 0.028 | 0.003 |
| sex | 0.015 | 0.001 | 0.015 | 0.002 | 0.014 | 0.000 | 0.011 | -0.002 | 0.005 | -0.008 | 0.020 | 0.007 |
| looking | 0.015 | 0.001 | 0.018 | 0.005 | 0.015 | 0.001 | 0.014 | 0.001 | 0.004 | -0.009 | 0.015 | 0.002 |
| gay | 0.013 | 0.001 | 0.016 | 0.004 | 0.013 | 0.001 | 0.010 | -0.002 | 0.006 | -0.006 | 0.013 | 0.001 |
| man | 0.012 | 0.001 | 0.011 | 0.000 | 0.009 | -0.002 | 0.007 | -0.004 | 0.015 | 0.004 | 0.012 | 0.001 |
| good | 0.010 | 0.001 | 0.008 | -0.002 | 0.006 | -0.003 | 0.008 | -0.001 | 0.015 | 0.006 | 0.009 | -0.001 |
| woman | 0.010 | 0.001 | 0.011 | 0.002 | 0.010 | 0.001 | 0.008 | -0.001 | 0.006 | -0.003 | 0.011 | 0.002 |
| mature | 0.010 | 0.001 | 0.020 | 0.010 | 0.007 | -0.003 | 0.006 | -0.003 | 0.001 | -0.009 | 0.014 | 0.004 |

Figure 25: Most skewed associations in FineWeb for 'christian' compared to other religions, measured using TF-IDF. Columns are sorted by the 'christian+' column, measuring the difference from the mean over all words.

| word | muslim | muslim+ | christian | christian+ | jewish | jewish+ | hindu | hindu+ | buddhist | buddhist+ | atheist | atheist+ |
|---|---|---|---|---|---|---|---|---|---|---|---|---|
| muslim | 0.115 | 0.083 | 0.011 | -0.021 | 0.018 | -0.015 | 0.027 | -0.006 | 0.015 | -0.017 | 0.009 | -0.024 |
| dating | 0.164 | 0.021 | 0.192 | 0.049 | 0.212 | 0.069 | 0.146 | 0.004 | 0.133 | -0.009 | 0.009 | -0.134 |
| women | 0.048 | 0.007 | 0.037 | -0.004 | 0.050 | 0.009 | 0.053 | 0.012 | 0.046 | 0.006 | 0.010 | -0.030 |
| sex | 0.020 | 0.007 | 0.015 | 0.001 | 0.015 | 0.002 | 0.014 | 0.000 | 0.011 | -0.002 | 0.005 | -0.008 |
| online | 0.043 | 0.006 | 0.047 | 0.010 | 0.057 | 0.020 | 0.038 | 0.001 | 0.032 | -0.005 | 0.004 | -0.033 |
| girl | 0.015 | 0.005 | 0.009 | -0.000 | 0.011 | 0.002 | 0.010 | 0.001 | 0.007 | -0.002 | 0.003 | -0.006 |
| girls | 0.016 | 0.005 | 0.012 | 0.000 | 0.016 | 0.004 | 0.014 | 0.002 | 0.010 | -0.002 | 0.003 | -0.009 |
| sites | 0.019 | 0.004 | 0.023 | 0.009 | 0.022 | 0.008 | 0.013 | -0.002 | 0.009 | -0.005 | 0.002 | -0.013 |
| mature | 0.014 | 0.004 | 0.010 | 0.001 | 0.020 | 0.010 | 0.007 | -0.003 | 0.006 | -0.003 | 0.001 | -0.009 |
| meet | 0.028 | 0.003 | 0.026 | 0.001 | 0.034 | 0.010 | 0.027 | 0.003 | 0.028 | 0.004 | 0.003 | -0.021 |
| woman | 0.011 | 0.002 | 0.010 | 0.001 | 0.011 | 0.002 | 0.010 | 0.001 | 0.008 | -0.001 | 0.006 | -0.003 |
| asian | 0.011 | 0.002 | 0.009 | -0.001 | 0.012 | 0.002 | 0.014 | 0.004 | 0.011 | 0.001 | 0.001 | -0.009 |
| free | 0.034 | 0.002 | 0.038 | 0.005 | 0.042 | 0.009 | 0.038 | 0.005 | 0.035 | 0.002 | 0.008 | -0.024 |
| site | 0.033 | 0.002 | 0.035 | 0.004 | 0.043 | 0.012 | 0.035 | 0.003 | 0.038 | 0.007 | 0.004 | -0.027 |
| looking | 0.015 | 0.002 | 0.015 | 0.001 | 0.018 | 0.005 | 0.015 | 0.001 | 0.014 | 0.001 | 0.004 | -0.009 |
| gay | 0.013 | 0.001 | 0.013 | 0.001 | 0.016 | 0.004 | 0.013 | 0.001 | 0.010 | -0.002 | 0.006 | -0.006 |
| best | 0.014 | 0.001 | 0.015 | 0.002 | 0.016 | 0.003 | 0.014 | 0.001 | 0.013 | -0.000 | 0.006 | -0.006 |
| men | 0.030 | 0.001 | 0.028 | -0.002 | 0.034 | 0.004 | 0.040 | 0.011 | 0.036 | 0.007 | 0.009 | -0.021 |
| man | 0.012 | 0.001 | 0.012 | 0.001 | 0.011 | 0.000 | 0.009 | -0.002 | 0.007 | -0.004 | 0.015 | 0.004 |

Figure 26: Most skewed associations in FineWeb for 'muslim' compared to other religions, measured using TF-IDF. Columns are sorted by the 'muslim+' column, measuring the difference from the mean over all words.

| word | jewish | jewish+ | hindu | hindu+ | buddhist | buddhist+ | atheist | atheist+ | muslim | muslim+ | christian | christian+ |
|---|---|---|---|---|---|---|---|---|---|---|---|---|
| jewish | 0.128 | 0.097 | 0.018 | -0.014 | 0.013 | -0.019 | 0.007 | -0.024 | 0.012 | -0.020 | 0.012 | -0.020 |
| dating | 0.212 | 0.069 | 0.146 | 0.004 | 0.133 | -0.009 | 0.009 | -0.134 | 0.164 | 0.021 | 0.192 | 0.049 |
| singles | 0.110 | 0.041 | 0.079 | 0.010 | 0.085 | 0.015 | 0.002 | -0.068 | 0.065 | -0.005 | 0.076 | 0.006 |
| online | 0.057 | 0.020 | 0.038 | 0.001 | 0.032 | -0.005 | 0.004 | -0.033 | 0.043 | 0.006 | 0.047 | 0.010 |
| site | 0.043 | 0.012 | 0.035 | 0.003 | 0.038 | 0.007 | 0.004 | -0.027 | 0.033 | 0.002 | 0.035 | 0.004 |
| mature | 0.020 | 0.010 | 0.007 | -0.003 | 0.006 | -0.003 | 0.001 | -0.009 | 0.014 | 0.004 | 0.010 | 0.001 |
| meet | 0.034 | 0.010 | 0.027 | 0.003 | 0.028 | 0.004 | 0.003 | -0.021 | 0.028 | 0.003 | 0.026 | 0.001 |
| personals | 0.032 | 0.009 | 0.031 | 0.009 | 0.034 | 0.012 | 0.001 | -0.021 | 0.018 | -0.004 | 0.018 | -0.004 |
| free | 0.042 | 0.009 | 0.038 | 0.005 | 0.035 | 0.002 | 0.008 | -0.024 | 0.034 | 0.002 | 0.038 | 0.005 |
| women | 0.050 | 0.009 | 0.053 | 0.012 | 0.046 | 0.006 | 0.010 | -0.030 | 0.048 | 0.007 | 0.037 | -0.004 |
| sites | 0.022 | 0.008 | 0.013 | -0.002 | 0.009 | -0.005 | 0.002 | -0.013 | 0.019 | 0.004 | 0.023 | 0.009 |
| single | 0.045 | 0.007 | 0.054 | 0.017 | 0.055 | 0.018 | 0.003 | -0.034 | 0.034 | -0.003 | 0.033 | -0.004 |
| looking | 0.018 | 0.005 | 0.015 | 0.001 | 0.014 | 0.001 | 0.004 | -0.009 | 0.015 | 0.002 | 0.015 | 0.001 |
| men | 0.034 | 0.004 | 0.040 | 0.011 | 0.036 | 0.007 | 0.009 | -0.021 | 0.030 | 0.001 | 0.028 | -0.002 |
| girls | 0.016 | 0.004 | 0.014 | 0.002 | 0.010 | -0.002 | 0.003 | -0.009 | 0.016 | 0.005 | 0.012 | 0.000 |
| gay | 0.016 | 0.004 | 0.013 | 0.001 | 0.010 | -0.002 | 0.006 | -0.006 | 0.013 | 0.001 | 0.013 | 0.001 |
| best | 0.016 | 0.003 | 0.014 | 0.001 | 0.013 | -0.000 | 0.006 | -0.006 | 0.014 | 0.001 | 0.015 | 0.002 |
| catholic | 0.014 | 0.003 | 0.006 | -0.004 | 0.010 | -0.001 | 0.012 | 0.002 | 0.006 | -0.004 | 0.015 | 0.005 |
| new | 0.016 | 0.002 | 0.013 | -0.001 | 0.013 | -0.001 | 0.014 | -0.000 | 0.013 | -0.001 | 0.014 | 0.000 |
| date | 0.012 | 0.002 | 0.012 | 0.002 | 0.011 | 0.002 | 0.002 | -0.008 | 0.010 | 0.000 | 0.012 | 0.002 |
| asian | 0.012 | 0.002 | 0.014 | 0.004 | 0.011 | 0.001 | 0.001 | -0.009 | 0.011 | 0.002 | 0.009 | -0.001 |
| girl | 0.011 | 0.002 | 0.010 | 0.001 | 0.007 | -0.002 | 0.003 | -0.006 | 0.015 | 0.005 | 0.009 | -0.000 |
| sex | 0.015 | 0.002 | 0.014 | 0.000 | 0.011 | -0.002 | 0.005 | -0.008 | 0.020 | 0.007 | 0.015 | 0.001 |
| chat | 0.016 | 0.002 | 0.017 | 0.003 | 0.019 | 0.005 | 0.001 | -0.013 | 0.014 | 0.000 | 0.016 | 0.002 |
| 100 | 0.011 | 0.002 | 0.011 | 0.002 | 0.013 | 0.003 | 0.002 | -0.007 | 0.008 | -0.001 | 0.010 | 0.000 |
| woman | 0.011 | 0.002 | 0.010 | 0.001 | 0.008 | -0.001 | 0.006 | -0.003 | 0.011 | 0.002 | 0.010 | 0.001 |
| essay | 0.010 | 0.001 | 0.017 | 0.007 | 0.013 | 0.003 | 0.003 | -0.007 | 0.007 | -0.003 | 0.009 | -0.001 |

Figure 27: Most skewed associations in FineWeb for 'jewish' compared to other religions, measured using TF-IDF. Columns are sorted by the 'jewish+' column, measuring the difference from the mean over all words.

| word | hindu | hindu+ | buddhist | buddhist+ | atheist | atheist+ | muslim | muslim+ | christian | christian+ | jewish | jewish+ |
|---|---|---|---|---|---|---|---|---|---|---|---|---|
| hindu | 0.129 | 0.100 | 0.020 | -0.009 | 0.002 | -0.027 | 0.015 | -0.015 | 0.004 | -0.026 | 0.006 | -0.023 |
| indian | 0.037 | 0.026 | 0.008 | -0.003 | 0.001 | -0.009 | 0.011 | 0.000 | 0.004 | -0.007 | 0.004 | -0.007 |
| single | 0.054 | 0.017 | 0.055 | 0.018 | 0.003 | -0.034 | 0.034 | -0.003 | 0.033 | -0.004 | 0.045 | 0.007 |
| women | 0.053 | 0.012 | 0.046 | 0.006 | 0.010 | -0.030 | 0.048 | 0.007 | 0.037 | -0.004 | 0.050 | 0.009 |
| men | 0.040 | 0.011 | 0.036 | 0.007 | 0.009 | -0.021 | 0.030 | 0.001 | 0.028 | -0.002 | 0.034 | 0.004 |
| singles | 0.079 | 0.010 | 0.085 | 0.015 | 0.002 | -0.068 | 0.065 | -0.005 | 0.076 | 0.006 | 0.110 | 0.041 |
| personals | 0.031 | 0.009 | 0.034 | 0.012 | 0.001 | -0.021 | 0.018 | -0.004 | 0.018 | -0.004 | 0.032 | 0.009 |
| essay | 0.017 | 0.007 | 0.013 | 0.003 | 0.003 | -0.007 | 0.007 | -0.003 | 0.009 | -0.001 | 0.010 | 0.001 |
| free | 0.038 | 0.005 | 0.035 | 0.002 | 0.008 | -0.024 | 0.034 | 0.002 | 0.038 | 0.005 | 0.042 | 0.009 |
| asian | 0.014 | 0.004 | 0.011 | 0.001 | 0.001 | -0.009 | 0.011 | 0.002 | 0.009 | -0.001 | 0.012 | 0.002 |
| dating | 0.146 | 0.004 | 0.133 | -0.009 | 0.009 | -0.134 | 0.164 | 0.021 | 0.192 | 0.049 | 0.212 | 0.069 |
| chat | 0.017 | 0.003 | 0.019 | 0.005 | 0.001 | -0.013 | 0.014 | 0.000 | 0.016 | 0.002 | 0.016 | 0.002 |
| site | 0.035 | 0.003 | 0.038 | 0.007 | 0.004 | -0.027 | 0.033 | 0.002 | 0.035 | 0.004 | 0.043 | 0.012 |
| meet | 0.027 | 0.003 | 0.028 | 0.004 | 0.003 | -0.021 | 0.028 | 0.003 | 0.026 | 0.001 | 0.034 | 0.010 |
| date | 0.012 | 0.002 | 0.011 | 0.002 | 0.002 | -0.008 | 0.010 | 0.000 | 0.012 | 0.002 | 0.012 | 0.002 |
| 100 | 0.011 | 0.002 | 0.013 | 0.003 | 0.002 | -0.007 | 0.008 | -0.001 | 0.010 | 0.000 | 0.011 | 0.002 |
| girls | 0.014 | 0.002 | 0.010 | -0.002 | 0.003 | -0.009 | 0.016 | 0.005 | 0.012 | 0.000 | 0.016 | 0.004 |
| gay | 0.013 | 0.001 | 0.010 | -0.002 | 0.006 | -0.006 | 0.013 | 0.001 | 0.013 | 0.001 | 0.016 | 0.004 |
| love | 0.016 | 0.001 | 0.014 | -0.001 | 0.013 | -0.002 | 0.013 | -0.001 | 0.017 | 0.003 | 0.014 | -0.000 |
| looking | 0.015 | 0.001 | 0.014 | 0.001 | 0.004 | -0.009 | 0.015 | 0.002 | 0.015 | 0.001 | 0.018 | 0.005 |
| girl | 0.010 | 0.001 | 0.007 | -0.002 | 0.003 | -0.006 | 0.015 | 0.005 | 0.009 | -0.000 | 0.011 | 0.002 |
| online | 0.038 | 0.001 | 0.032 | -0.005 | 0.004 | -0.033 | 0.043 | 0.006 | 0.047 | 0.010 | 0.057 | 0.020 |
| best | 0.014 | 0.001 | 0.013 | -0.000 | 0.006 | -0.006 | 0.014 | 0.001 | 0.015 | 0.002 | 0.016 | 0.003 |
| woman | 0.010 | 0.001 | 0.008 | -0.001 | 0.006 | -0.003 | 0.011 | 0.002 | 0.010 | 0.001 | 0.011 | 0.002 |

Figure 28: Most skewed associations in FineWeb for 'hindu' compared to other religions, measured using TF-IDF. Columns are sorted by the 'hindu+' column, measuring the difference from the mean over all words.

**A**

| word | old | old+ | young | young+ |
| --- | --- | --- | --- | --- |
| old | 0.034 | 0.012 | 0.010 | -0.012 |
| just | 0.019 | 0.002 | 0.016 | -0.002 |
| new | 0.018 | 0.002 | 0.015 | -0.002 |
| like | 0.021 | 0.002 | 0.017 | -0.002 |
| ve | 0.011 | 0.001 | 0.009 | -0.001 |
| don | 0.013 | 0.001 | 0.010 | -0.001 |
| ll | 0.009 | 0.001 | 0.006 | -0.001 |
| use | 0.009 | 0.001 | 0.006 | -0.001 |
| time | 0.019 | 0.001 | 0.017 | -0.001 |
| really | 0.012 | 0.001 | 0.009 | -0.001 |
| good | 0.013 | 0.001 | 0.011 | -0.001 |
| little | 0.010 | 0.001 | 0.008 | -0.001 |
| got | 0.009 | 0.001 | 0.007 | -0.001 |
| things | 0.010 | 0.001 | 0.008 | -0.001 |
| know | 0.013 | 0.001 | 0.011 | -0.001 |
| make | 0.012 | 0.001 | 0.011 | -0.001 |
| want | 0.010 | 0.001 | 0.009 | -0.001 |
| look | 0.008 | 0.001 | 0.007 | -0.001 |
| need | 0.009 | 0.001 | 0.008 | -0.001 |
| home | 0.010 | 0.001 | 0.009 | -0.001 |
| right | 0.009 | 0.001 | 0.008 | -0.001 |
| going | 0.010 | 0.001 | 0.009 | -0.001 |
| day | 0.012 | 0.001 | 0.011 | -0.001 |
| way | 0.012 | 0.001 | 0.011 | -0.001 |
| great | 0.010 | 0.001 | 0.009 | -0.001 |
| think | 0.011 | 0.001 | 0.010 | -0.001 |

**B**

| word | young | young+ | old | old+ |
| --- | --- | --- | --- | --- |
| young | 0.038 | 0.016 | 0.006 | -0.016 |
| children | 0.016 | 0.005 | 0.007 | -0.005 |
| women | 0.011 | 0.003 | 0.006 | -0.003 |
| school | 0.013 | 0.003 | 0.008 | -0.003 |
| said | 0.017 | 0.002 | 0.012 | -0.002 |
| people | 0.021 | 0.002 | 0.016 | -0.002 |
| child | 0.009 | 0.002 | 0.005 | -0.002 |
| life | 0.015 | 0.001 | 0.012 | -0.001 |
| family | 0.011 | 0.001 | 0.008 | -0.001 |
| story | 0.009 | 0.001 | 0.007 | -0.001 |
| world | 0.012 | 0.001 | 0.010 | -0.001 |
| man | 0.010 | 0.001 | 0.008 | -0.001 |
| book | 0.010 | 0.001 | 0.008 | -0.001 |

Figure 29: *Age* bias in FineWeb, measured as most skewed associations for 'old' and 'young', using TF-IDF. Sorted by the difference from the mean TF-IDF for all words associated to 'old' ('old+', A) and 'young' ('young+', B).

