# OpenReview forum: "The FineWeb Datasets: Decanting the Web for the Finest Text Data at Scale"
_NeurIPS.cc/2024/Datasets_and_Benchmarks_Track — NeurIPS 2024 Track Datasets and Benchmarks Spotlight_

### Official Review · Reviewer_RcqR · 2024-06-15
**Thanks for your work!**

**Rating:** 7
**Confidence:** 4
**Correctness:** Everything seems correct.
**Clarity:** The writing is easy to understand.

**Review:**

This paper provides a valuable service to the community.  The dataset has a more permissive license than, for example, RefinedWeb which will benefit many.  The cost of constructing the dataset and optimizing the techniques that went into this are prohibitively high for all but few, so it’s nice that people with the resources made this public.

In general, my biggest criticism of this paper is that I didn’t learn much from it.  It reads more as a tech report than a paper. If I want to build an even better dataset, I don’t really know where to start aside from just trying a lot of stuff and seeing which produces the best models on benchmarks.

I believe this paper fits squarely within the purview of the datasets and benchmarks track.

See additional strengths and weaknesses below.

**Strengths:**

One thing I liked about the paper is the narrativization of the dataset curation process.  The reader can see exactly what work went into building FineWeb.  Many other papers instead jump to the finished product.

The paper is also very thorough, and the writing is easy to understand.

As I mentioned above, this dataset provides a service to many in the community, and I appreciate the open science.

This work makes the interesting observation that good text is more likely to be duplicated than bad text, much in the way that there are far more ways to be wrong than to be right so wrong stuff is less likely to be repeated.  It could be valuable to explore this observation more.

**Additional Feedback:**

N/A

**Documentation:**

Everything is fairly well documented, and they link to the actual datasets.

**Limitations:**

The authors do discuss limitations sufficiently in the final section of the paper.

**Opportunities For Improvement:**

As stated above, my biggest complaint is that I didn’t really learn much from this paper.  I don’t know why different design decisions have a positive impact, just that they do.

It would be valuable to know how important it is to have clean data for most of training.  Can we train on lots of junk and then do a cool-down on, e.g. FineWeb, for late stage training?

I appreciate that the authors discuss bias, but I also wonder how data filtering skews the types or topics of documents in the pretraining data.  Are some kinds of data more likely to get filtered out?  Does this create a skewed data distribution that impacts the model?  If we understand that, can we de-bias the models with the right data mix?

The downstream benchmark scores are great, but does cleaning the data have a negative impact on the diversity of generated content?

**Relation To Prior Work:**

The authors to a great job of discussing the data curation pipelines of previous pretraining datasets and mention when their procedure is similar or deviates.

**Summary And Contributions:**

This paper introduces a new LLM pretraining dataset, FineWeb, as well as a more filtered version, FineWeb-Edu, that contains educational content.  The authors ablate away their design choices with smaller LLMs and show how each choice impacted performance on a suite of downstream benchmark tasks.  Along the way, the authors make interesting observations about deduplication.

---

> ### Author Rebuttal · Authors · 2024-08-14
>
> We thank reviewer `RcqR` for their review and for the interesting hypothetical questions regarding the theoretical basis for why some of our empirically validated design decisions have a positive impact.
> We appreciate the reviewer’s points regarding topic diversity, and will add a section to the appendix comparing the topic distribution, as well as domain fit, between FineWeb and FineWeb-Edu.

---

### Official Review · Reviewer_DjLJ · 2024-07-18
**Overall a significant and highly valuable dataset, with some room for improvement on the evaluation setup.**

**Rating:** 7
**Confidence:** 5
**Correctness:** I believe the claims to be correct du…

**Review:**

### Summary of Review
The paper is clearly written, adheres to scientific standards is original in nature. The contribution is highly significant as the work presents openly accessible datasets which are among the largest pretraining datasets. Thanks to its openness, it is thus likely that this work will facilitate and accelerate the development of LLMs both for research and industry applications. The authors also provide significant evidence in the form of ablation models, supporting the claim on quality of the released data.

Nevertheless, I see some room for improvement. My main concern with the presented work and the methodology is on the evaluation protocol. Namely, the choice of benchmarks, based on which the data curation strategies is chosen, appears to be relatively narrow. Did the authors consider other benchmarks? Why did the authors not evaluate perplexity on different domains? How would the evaluation scores change if benchmarks were aggregated not as an average of raw scores, but of normalized scores? I think the evaluation protocol in the paper would benefit from the following three points:
1. A broader evaluation scheme such as is done in eg [1].
2. Evaluations on language model fit: perplexity on different (eg on the domains chosen in Paloma [2])
3. An evaluation of scores aggregated as either an average over normalized scores, or as a sum of ranks.

### Pros:
- among the largest open and high quality web dataset (15T tokens)
- presents innovative and new approaches to filter the dataset for educational content
- datasets are shown to produce strong models on the considered benchmarks, outperforming other datasets on the considered benchmark tasks
- step-by-step explanations of curation rationales

### Cons:
- choice of benchmarks is relatively narrow, and evaluation should be conducted more holistically
- the aggregation of individual benchmarks as an average over raw scores can be somewhat misleading because of the different scales of individual benchmarks.


### References
[1] Li et al, DataComp-LM: In search of the next generation of training sets for language models, 2024.

[2] Magnusson et al., Paloma: A Benchmark for Evaluating Language Model Fit, 2023.

**Strengths:**

- One of the largest open and high quality web datasets (15T tokens),
- open sourcing tools for data curation, model training and evaluation
- presents innovative and new approaches to filter the dataset for educational content
- datasets are shown to produce strong models, outperforming other datasets on the considered benchmark tasks
- presents detailed analysis with ablations, and step-by-step explanations of curation rationales.

**Additional Feedback:**

To create the FineWeb-Edu dataset, the authors used 3 as a threshold to filter the dataset because higher thresholds degraded the performance. Can the authors elaborate on this finding a bit more? Why does a lower quality threshold lead to a better model?

**Clarity:**

The paper is generally clearly written and provides sufficient details to follow.

### Baselines
In Figure 9 and Section 3.7, where the FineWeb datasets are compared to baselines, it needs to be stated more clearly that the RedPajama2 baseline is an unfiltered dataset.

### Deduplication
In the main part of the paper, the authors explain that:
```
"(L183/184) Documents with the same 8 MinHashes in any bucket are considered duplicates of each other."
```
However, in MinhashLSH, after the initial (groupby) step on individual buckets, there is typically a subsequent connected components step where a cluster of duplicates is formed by grouping all documents in a transitive way, based on the similarity relation from the first bucketing step: if A ~ B and B ~ C, then A ~ C even if the minhashes of A and C are different in all of the buckets. Is this step part of the deduplication pipeline? I think this needs to be stated more clearly in the paper.

**Documentation:**

The paper, together with the released data curation pipeline, training and evaluation framework provide sufficient documentation.

**Ethics:**

Ethical considerations are discussed at a satisfactory level in Section 5 and in the datasheet.

**Limitations:**

- mixing with other sources not explored, although this is stated clearly in the paper. I do not see this as a significant limitation.
- narrow evaluation setup
- generation capabilities of resulting models is not explored (eg, eval on squad, coqa). This limitation stems from the relatively narrow evaluation setup.

**Opportunities For Improvement:**

- For reproducibility, the authors should consider publishing the full set of hyperparameters used to train the ablation models.
- The paper presents an aggregate score based on an averaging 8 benchmarks. Since these benchmarks have accuracy on different scales (e.g., 25% random baseline on MMLU vs 50% on PIQA), this aggregation can potentially distort the findings. Presenting a more holistic evaluation scheme as mentioned above would give the work additional strength.
- More statistics on the dataset composition would further strengthen the paper. Next to the presented analysis on biases, I would especially like to see a distribution over topics in the datasets and how this distribution differs between fineweb vs. fineweb-edu.

**Relation To Prior Work:**

Relation to prior work is discussed to enough detail and section 2 gives the reader ample background to situate the current work in the body of relevant literature.

**Summary And Contributions:**

This paper introduces the FineWeb datasets, a series of web-based datasets with up to 15 trillions token derived from 96 Common Crawl snapshots, designed to train large language models. The paper presents detailed ablation studies on different techniques to filter and deduplicate web documents and results in increasingly better datasets, measured in terms of accuracy score averaged over a selection of 8 academic benchmarks. Next to the exploration of existing filters, this work also presents new strategies to filter and deduplicate datasets. It is shown that the resulting datasets, FineWeb and FineWeb-Edu outperform other datasets including C4 and RefinedWeb on the benchmarks considered.

---

> ### Author Rebuttal · Authors · 2024-08-14
>
> We would like to thank reviewer `DjLJ` for their detailed review and extensive feedback regarding the evaluation setup. We address the points raised by the reviewer below.
>
> ## Evaluations
>
> > Did the authors consider other benchmarks? Why did the authors not evaluate perplexity on different domains? How would the evaluation scores change if benchmarks were aggregated not as an average of raw scores, but of normalized scores? I think the evaluation protocol in the paper would benefit from the following three points:
> >
> > 1. A broader evaluation scheme such as is done in eg \[1\].
> > 2. Evaluations on language model fit: perplexity on different (eg on the domains chosen in Paloma \[2\])
> > 3. An evaluation of scores aggregated as either an average over normalized scores, or as a sum of ranks.
> > generation capabilities of resulting models is not explored (eg, eval on squad, coqa). This limitation stems from the relatively narrow evaluation setup.
>
> We considered a large number of benchmark tasks but, unfortunately, most of them provide little to no signal at the small model scale we used for our experiments. Due to computational constraints, we trained our models on a relatively small (\<2B params) scale and selected benchmark tasks whose scores showed small variance between different data sampling and training seeds of the same dataset; monotonic behavior (scores increasing over training without large oscillations); and were significantly above random baseline. We also decided to run evaluations in a 0-shot setting, as few-shot may introduce additional confounders, such as the choice and answer distribution of the few-shot examples. We note that many of the tasks from Li et al (2024, “DataComp-LM”) are not in the 0-shot setting. We agree with the limitation identified by the reviewer regarding the lack of generative tasks such as squad in our group of benchmark tasks.
>
> While multiple domains of Paloma are based on parts of some of the datasets we compare our dataset with (such as C4, Dolma, RefinedWeb, etc), and thus possibly not well suited for a comparison with these datasets, we see the reviewer’s point regarding measuring domain fit, and will add a comparison between FineWeb and FineWeb-Edu (also related to the reviewer’s point regarding topic distribution between these two versions).
>
> We also agree with the reviewer when it comes to using normalized task scores for score aggregation, as done in other recent work. We have confirmed that using a normalized score does not change the relative ordering of the datasets in our comparison group and will add a note in the paper itself.
>
> ## Clarifications
>
> We will clarify the RedPajama note, as well as make it explicit that we also perform the standard clustering step in Minhash deduplication.
>
> ## Further analysis
>
> > More statistics on the dataset composition would further strengthen the paper. Next to the presented analysis on biases, I would especially like to see a distribution over topics in the datasets and how this distribution differs between fineweb vs. fineweb-edu.
>
> We thank the reviewer for this suggestion, which we will add, together with the domain fit comparison (as mentioned above).
>
> ## FineWeb-Edu
>
> > To create the FineWeb-Edu dataset, the authors used 3 as a threshold to filter the dataset because higher thresholds degraded the performance. Can the authors elaborate on this finding a bit more? Why does a lower quality threshold lead to a better model?
>
> Filtering for samples with an educational score of 4 or more would only keep 1.14% of the original FineWeb (as opposed to \~8% with a score of 3 or more). When training on this data we observe degraded performance compared to the data with a score of 3 or more. We believe that the higher threshold might be too strict and lead to lack of diversity in the training data, which may, in turn, harm generalization capabilities.
>
> ## Misc
>
> > For reproducibility, the authors should consider publishing the full set of hyperparameters used to train the ablation models.
>
> We will update the paper with this information.

---

### Official Review · Reviewer_rXCs · 2024-07-25
**FineWeb Review**

**Rating:** 8
**Confidence:** 5
**Correctness:** The paper is about a dataset and the …
**Clarity:** The paper is well written.

**Review:**

**quality and clarity:**
The paper is of high quality and is a good read, as it carefully documents and explains all design choices.

**originality and significance:**
The FineWeb dataset represents a big step in the era of LLMs. While there are other open-source datasets, FineWeb is the largest and of higher quality (except for concurrent work on DataComp-LM https://arxiv.org/abs/2406.11794).

**pros:**
- **comprehensive documentation:** The detailed explanation of design choices and ablation studies enhances the reproducibility and credibility of the dataset.
- **large scale:** The extensive size of the FineWeb dataset is likely to provide robust training data for large-scale language models.
- **open-source availability:** Making the dataset and pipelines open-source supports further research and development.
- **performance improvement:** The dataset shows improvements in benchmarks like MMLU and ARC, validating its effectiveness.

**cons:**
- **snapshot timeliness:** I concern and suspect that the most recent Common Crawl snapshots used in FineWeb compasion to other datasets, so these snapshots might be more contaminated with evaluation data compared to older snapshots.
- **comparison of snapshots:** The paper may want to compare different snapshots of FineWeb, which could offer insights into the dataset's evolution and its impact on model performance.
- **filtering methodology for fineweb-edu:** The filtering methodology for FineWeb-Edu could be compared with other legacy approaches (as noted in the limitations section).
- **licensing concerns:** The use of Llama 3 for creating a classfier that creates FineWeb-Edu might raise licensing issues (as mentioned in the limitations section).

**Strengths:**

- The paper carefully documents and ablates all of the design choices used in FineWeb.
- FineWeb achieves the best overall performance in the open-source English datasets.

**Additional Feedback:**

I have plotted an image of what I sometimes think about all filtered datasets. This could happen in the long run (maybe after 15 trillion tokens); I do not have any proof, but of course, not everyone has the resources to test this out (and use it). So, every filtered dataset is useful in some way or another.

[FineWeb.jpg](https://postimg.cc/yJCD5X2Q)

**Documentation:**

The dataset includes sufficient information on how the data was collected, by whom, and from where. A URL for access is also provided.

**Ethics:**

No ethical concerns, although it is not possible to ensure that FineWeb does not contain any personal/private information. The FineWeb replaces emails and IPs, but not the rest. For example, extending this set to phone numbers requires an accurate regex, which is not always reliable.

**Limitations:**

**FineWeb-Edu**:

I have some concerns about the data being filtered from the FineWeb-Edu. Perhaps showing it's performance on some other benchmarks here would be wiser to highlight aspects that this educational data might be missing.

FineWeb-Edu is a bit of an overkill; using a simpler method in comparison would also be worth considering. For example, Brown et al. (http://arxiv.org/abs/2005.14165) use a filter based on a binary logistic regression classifier trained with n-gram features to distinguish between reference corpora (Books3, Wikipedia, and OpenWebText) and a random sample of Common Crawl. The paper has this sort of comparison for FineWeb but for FineWeb-Edu the design choices are not shown.

Also, Llama 3 (not Llama 3.1) prohibits using the Llama Materials or outputs to improve any other large language model, except for Meta Llama 3 or its derivatives. Does this mean that using Llama-3-70B-Instruct to create a classifier that generates FineWeb-Edu would restrict its license? Currently, the same license as FineWeb is used for FineWeb-Edu.

**Opportunities For Improvement:**

- In Figure 9, when comparing FineWeb datasets to other public datasets, did the authors use the part of FineWeb from the most recent Common Crawl snapshots? This is because the other datasets are related to earlier snapshots, and there is a possibility that the most recent snapshots might be more contaminated with evaluation data.

- Any comparison between different shanpshots?

**Relation To Prior Work:**

The paper discusses and directly compares with related works.

**Summary And Contributions:**

The paper presents the FineWeb Dataset, a 15-trillion token dataset derived from 96 Common Crawl snapshots (almost all of the Common Crawl snapshots). They carefully document and ablate all of the design choices used in FineWeb, which requires extensive GPU hours to work out. They also introduce FineWeb-Edu, a 1.3-trillion token collection of educational text filtered from FineWeb. Models trained on FineWeb-Edu show improved performance on knowledge and reasoning-intensive benchmarks such as MMLU and ARC. The pipelines, ablation models, and data are available as open-source.

---

> ### Author Rebuttal · Authors · 2024-08-14
>
> We thank reviewer `rXCs` for their thorough review. We appreciate the relevant questions raised about different commoncrawl snapshots providing different performance benefits, as well as the points about FineWeb-Edu. We address the concerns raised by the reviewer below.
>
> ## CommonCrawl snapshots
>
> > * **snapshot timeliness:** I concern and suspect that the most recent Common Crawl snapshots used in FineWeb compasion to other datasets, so these snapshots might be more contaminated with evaluation data compared to older snapshots.
> > * **comparison of snapshots:** The paper may want to compare different snapshots of FineWeb, which could offer insights into the dataset's evolution and its impact on model performance.
> > * In Figure 9, when comparing FineWeb datasets to other public datasets, did the authors use the part of FineWeb from the most recent Common Crawl snapshots? This is because the other datasets are related to earlier snapshots, and there is a possibility that the most recent snapshots might be more contaminated with evaluation data.
>
> For our dataset comparisons, we randomly sampled the training tokens from the entire dataset, meaning that indeed some data from the most recent snapshots was included, but so was data from all other snapshots (as we processed all snapshots that existed at the time). We did not artificially upsample the newer snapshots.
>
> We share the reviewer’s concerns regarding differences in commoncrawl snapshots. Early on in the dataset creation process (after we had performed per snapshot deduplication of the data), we trained models on each individual snapshot. This study revealed that there is an upward trend in the “quality” of snapshots over time, but we could not find a clear correlation between this performance increase and changes in the amount of benchmark data contamination present in the dataset (we checked for n-gram overlap between the dataset and the benchmark questions/answers). Importantly, the FineWeb data used for comparison with other datasets also contains the earliest commoncrawl snapshots, which exhibit lower performance than those used by some of the other datasets we compared against.
>
> Roughly, our results showed that the 2017-2018 snapshots were significantly more performant than the earlier snapshots, and that from 2019 to the latest dumps there is a clear upwards trend in performance. We ultimately decided that this analysis was more about CommonCrawl itself and therefore out of the scope of the paper.
>
> We would also like to point out that in section 3.5, where we ablate different filters against C4, these ablations were based on the 2019-18 snapshot, the same that was used to create C4, in an effort to keep the comparison fair.
>
> ## FineWeb-Edu
>
> > I have some concerns about the data being filtered from the FineWeb-Edu. Perhaps showing it's performance on some other benchmarks here would be wiser to highlight aspects that this educational data might be missing.
> >
> > FineWeb-Edu is a bit of an overkill; using a simpler method in comparison would also be worth considering. For example, Brown et al. ([http://arxiv.org/abs/2005.14165](http://arxiv.org/abs/2005.14165)) use a filter based on a binary logistic regression classifier trained with n-gram features to distinguish between reference corpora (Books3, Wikipedia, and OpenWebText) and a random sample of Common Crawl. The paper has this sort of comparison for FineWeb but for FineWeb-Edu the design choices are not shown.
>
> We agree with the limitations identified by the reviewer regarding not documenting and comparing different classification approaches, and the concerns about potential drawbacks of the education classifier. We will add an additional analysis to the Appendix comparing the topic clusters on FineWeb and FineWeb-Edu (as also suggested by another reviewer).
>
> > Also, Llama 3 (not Llama 3.1) prohibits using the Llama Materials or outputs to improve any other large language model, except for Meta Llama 3 or its derivatives. Does this mean that using Llama-3-70B-Instruct to create a classifier that generates FineWeb-Edu would restrict its license? Currently, the same license as FineWeb is used for FineWeb-Edu.
>
> The data from FineWeb-Edu is a subset of the FineWeb data, with the Llama model only being used indirectly to filter the dataset, but in any case this is a legal question without a clear answer. We believe the entire process could be easily replicated with Llama 3.1, with its more permissive license, to similar or better results.
>
> ## Additional feedback
>
> > I have plotted an image of what I sometimes think about all filtered datasets. This could happen in the long run (maybe after 15 trillion tokens); I do not have any proof, but of course, not everyone has the resources to test this out (and use it). So, every filtered dataset is useful in some way or another.
>
> We find the hypothesis and plot made by the reviewer thought-provoking and have considered similar questions. Overall we would love to run all ablations and explorations at extremely large scale, but as you note it is cost-prohibitive to do so. Ultimately, we hope that the question of whether our filtering holds up at larger scales will be resolved by future work (by groups with more computational resources…).

---

> > ### Comment · Reviewer_rXCs · 2024-09-04
> > **Response by Reviewer**
> >
> > > We did not artificially upsample the newer snapshots.
> >
> > Glad to hear that.
> >
> > > We will add an additional analysis to the Appendix comparing the topic clusters on FineWeb and FineWeb-Edu (as also suggested by another reviewer).
> >
> > Thank you.
> >
> > > However, this is a legal question without a clear answer.
> >
> > I agree.
> >
> >  > Ultimately, we hope that the question of whether our filtering holds up at larger scales will be resolved by future work.
> >
> >  I hope so as well.
> >
> > Thank you so much for providing detailed answers to the questions raised; most of my concerns have been resolved. I did not factor most of these concerns, except those related to FineWeb-Edu, into the scoring, and a score of 8 (clear accept) reflects the paper's significant contribution.

---

### Official Review · Reviewer_gcLr · 2024-08-04
**Meaningful empirical contribution**

**Rating:** 7
**Confidence:** 3
**Correctness:** The claims seem correct.
**Clarity:** Yes

**Review:**

The paper is well written and easy to follow. While the paper does not introduce any significant technical ideas, it makes a meaningful empirical contribution. The paper also sheds light on construction of LLM pre-training corpora which is usually not published. The artifacts released in the paper are likely to enable future work.

**Strengths:**

- The paper is well written.
- The paper makes a valuable experimental contribution.
- The paper releases meaningful artifacts which will enable future work.

**Additional Feedback:**

- Please clarify how the ablations should be understood: For each ablation experiment sub-section in Section 3 what is the experimental condition for the unablated stages? For example, what is the deduplication strategy and other filtering when ablating text extraction?
- Please consider releasing the progressively processed versions of the dataset - this is likely to ease the engineering necessary to reproduce or build on your experiments.
- Similarly, consider annotating additional data from the data filtering labels in your release of the dataset - for example any outputs from filtering models for a document.
- Consider releasing the training set for the FineWeb-Edu classifier.
- What was the procedure for iterating on the Llama-3 prompt for labeling educational data? Were the 410,000 synthetic annotations on webpages or some other piece of text? Was this the result of applying the Llama-3 model on FineWeb in its entirity or only a part of it?

**Documentation:**

The dataset is well documented and the authors seem to release code for constructing the data.

**Opportunities For Improvement:**

- Some aspects of the experimental setup can be clarified.
- Additional details of the dataset release could be clarified.

**Relation To Prior Work:**

The paper discusses prior work well.

**Summary And Contributions:**

The paper described the FineWeb dataset for pretraining large language models. The dataset documents the text extraction, filtering, and deduplication strategies and ablates a design choices for each of these steps. The paper also describes FineWeb-Edu, a dataset filtered from the larger FineWeb to contain "educational" texts using a classifier trained to identify educational text on a synthetic training set. The value of this additionally filtered dataset is also demonstrsted in experiments.

---

> ### Author Rebuttal · Authors · 2024-08-14
>
> We thank reviewer `gcLr` for positively assessing our work, and for mentioning relevant points that can be made clearer in the paper. We will update our draft to reflect these concerns.
> We address the points raised by the reviewer below.
>
> ## Ablations
>
> > Please clarify how the ablations should be understood: For each ablation experiment sub-section in Section 3 what is the experimental condition for the unablated stages? For example, what is the deduplication strategy and other filtering when ablating text extraction?
>
> Ablations shown in each Section 3 subsection follow our iterative dataset building process, i.e., the baseline model for each section includes only the processing steps from the previous sections. Similarly, when different approaches are compared, they each have as their starting point the processing decisions validated in the preceding sections. For the text extraction experiment specifically, no deduplication or filtering is applied, except for the language filter (as we were only interested in English data).
>
> ## Artifacts and annotations
>
> > * Please consider releasing the progressively processed versions of the dataset \- this is likely to ease the engineering necessary to reproduce or build on your experiments.
> > * Similarly, consider annotating additional data from the data filtering labels in your release of the dataset \- for example any outputs from filtering models for a document.
>
> Our final released dataset comprises several terabytes of data. Given that each filtering stage removes a large amount of data, the intermediate versions of the full dataset added together are in the petabytes range and are therefore unfortunately challenging to release and host. We will, however, investigate the feasibility of releasing an extract/sample from each pipeline stage.
>
> Regarding annotations, we include a `language_score` field on each sample, and for FineWeb-Edu the released dataset also includes the educational score as determined by the classifier in the `score` field. These are the only models that we employed for filtering. We will make this clearer in the Appendix.
>
> > Consider releasing the training set for the FineWeb-Edu classifier.
>
> This dataset is available at [https://huggingface.co/datasets/HuggingFaceFW/fineweb-edu-llama3-annotations](https://huggingface.co/datasets/HuggingFaceFW/fineweb-edu-llama3-annotations). We will add a link to this repository in the Supplementary materials in the “Linked resources” section.
>
> ## Samples annotated for educational value
>
> > What was the procedure for iterating on the Llama-3 prompt for labeling educational data? Were the 410,000 synthetic annotations on webpages or some other piece of text? Was this the result of applying the Llama-3 model on FineWeb in its entirity or only a part of it?
>
> We refined the prompt by starting from Yuan et al.'s prompt for evaluating instruction data, then adapting it for rating the educational value of web pages. Our team iteratively reviewed Llama3's annotations, adjusting the prompt wording to ensure the scores aligned closely with human judgment, for example, by instructing the model to prioritize grade and middle school level content for higher scores rather than advanced academic material, which was initially overrepresented in the form of complex research paper abstracts.
>
> The 410,000 samples that were annotated by Llama-3 were randomly sampled webpages from the FineWeb *CC-MAIN-2024-10* snapshot. This is only a small part of the full dataset as classifying the entire dataset with Llama-3 directly would be prohibitively expensive. We subsequently used these annotations to train a (much smaller) classifier model, which in turn was used to classify the entire dataset.

---

### Decision · Program_Chairs · 2024-09-26

**Decision:**

Accept (Spotlight)

**Comment:**

This paper presents a large (15 trillion tokes) open publicly available high quality dataset for LLM pretraining. Importantly, it documents all the design choices in the preparation of the dataset from the Common Crawl. It also shows how each of these design choices impacts performance. Importantly, not just the dataset but the code to generate the dataset is released. In general, the code repository and dataset are well documented. There is some concern about the fact there is only one temporal snapshot of the data (although this is normally the case with pretraining datasets) as well as the fact that the evaluation tasks could have been expanded. That being said this is a resource that adds a lot to the community both in terms of our understanding and potential use for training large parameter open LLMs